# CoDeC: Communication-Efficient Decentralized Continual Learning

**Sakshi Choudhary**                                                     *choudh23@purdue.edu*
*Department of Electrical and Computer Engineering*
*Purdue University*

**Sai Aparna Aketi**                                                       *saketi@purdue.edu*
*Department of Electrical and Computer Engineering*
*Purdue University*

**Gobinda Saha**                                                            *gsaha@purdue.edu*
*Department of Electrical and Computer Engineering*
*Purdue University*

**Kaushik Roy**                                                            *kaushik@purdue.edu*
*Department of Electrical and Computer Engineering*
*Purdue University*

**Reviewed on OpenReview:** *https://openreview.net/forum?id=NO5OnQG1BA*

## Abstract

Training at the edge utilizes continuously evolving data generated at different locations. Privacy concerns prohibit the co-location of this spatially as well as temporally distributed data, deeming it crucial to design training algorithms that enable efficient continual learning over decentralized private data. Decentralized learning allows serverless training with spatially distributed data. A fundamental barrier in such setups is the high bandwidth cost of communicating model updates between agents. Moreover, existing works under this training paradigm are not inherently suitable for learning a temporal sequence of tasks while retaining the previously acquired knowledge. In this work, we propose CoDeC, a novel communication-efficient decentralized continual learning algorithm that addresses these challenges. We mitigate catastrophic forgetting while learning a distributed task sequence by incorporating orthogonal gradient projection within a gossip-based decentralized learning algorithm. Further, CoDeC includes a novel lossless communication compression scheme based on the gradient subspaces. We theoretically analyze the convergence rate for our algorithm and demonstrate through an extensive set of experiments that CoDeC successfully learns distributed continual tasks with minimal forgetting. The proposed compression scheme results in up to $4.8\times$ reduction in communication costs without any loss in performance. [1]

## 1 Introduction

Deep neural networks have demonstrated exceptional performance for many visual recognition tasks over the past decade. This has been fueled by the explosive growth of available training data and powerful computing resources. Edge devices such as smartphones, drones, and Internet-of-Things (IoT) sensors contribute towards generating this massive amount of data (Shi et al., 2020). Interestingly, this data is spatially distributed, while continuously evolving over time. Large-scale deep neural network training has traditionally relied upon the availability of a humongous amount of data at a central server. This mainly poses three

---

[1]The PyTorch implementation can be found at `https://github.com/Sakshi09Ch/CoDeC`

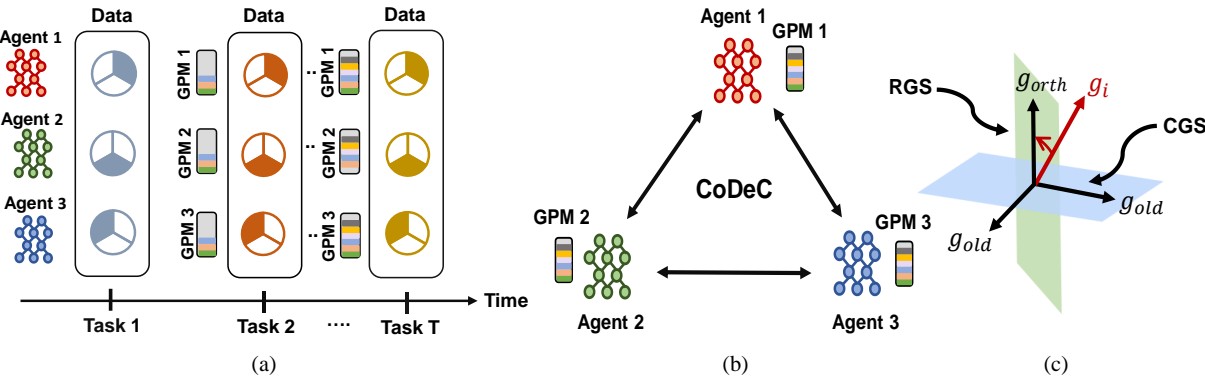

Figure 1: An overview of CoDeC. (a) Data for each incoming task is independently and identically distributed (IID) over the decentralized agents. Each agent has a GPM (Gradient Projection Memory) which is updated after learning each task. (b) Based on the sparse graph topology, the agents communicate coefficients associated with the model updates at each training iteration. (c) GPM partitions each layer's subspace into two orthogonal subspaces.

challenges: (1) high network bandwidth requirements to collect this dispersed data from numerous learning agents, (2) data privacy concerns for locally-generated data accessed by the central server and (3) adapting to changing data distributions without expensive training from scratch. This motivates the need for learning algorithms to enable efficient distributed training by utilizing spatially and temporally distributed data.

Centralized distributed learning (i.e. federated learning) has emerged to train models over spatially distributed data without compromising on user privacy Konečný et al. (2016). This approach relies upon a central parameter server to collect local model updates, process, and send the global updates back to the agents. However, the central server may lead to a single point of failure and network bandwidth issues (Assran et al., 2019). To address these concerns, several decentralized learning algorithms have been developed (Bianchi et al., 2013; Lan et al., 2017; Lian et al., 2017; Assran et al., 2019). Decentralized learning is a peer-to-peer learning paradigm, where agents communicate only with their neighbors without the need for a central server. Each agent learns a global generalized model by aggregating locally computed model updates shared by neighbors. However, decentralized learning algorithms are not inherently equipped to thrive in dynamic learning environments with a temporal sequence of changing data distributions.

Gradient-based optimization methods like plain SGD and DPSGD (Lian et al., 2017) inherently update model parameters by minimizing the loss function with respect to the current data distribution. This results in overwriting of parameters learned for the previous task(s), leading to catastrophic forgetting (Mccloskey & Cohen, 1989; Ratcliff, 1990). Hence, continual learning techniques focus on learning consecutive tasks without forgetting the past acquired knowledge (Lee et al., 2017; Lopez-Paz & Ranzato, 2017; Saha et al., 2021a; Mallya et al., 2018; Kirkpatrick et al., 2017; Farajtabar et al., 2020; Wang et al., 2021; Saha et al., 2021).

Table 1 summarizes previous works that address the challenges of learning with spatially and/or temporally distributed data. In this paper, we propose CoDeC to enable serverless training with data distributed across space as well as time. To the best of our knowledge, this is the first work that demonstrates such a decentralized continual learning setup. Our algorithm has three components: (1) SGD combined with gossip averaging (Xiao & Boyd, 2003) as shown in (Lian et al., 2017) to learn with spatially distributed private data,(2) Gradient Projection Memory (GPM) (Saha et al., 2021) to continually learn a temporal task sequence with minimal forgetting and (3) a novel lossless communication compression scheme to reduce the bandwidth requirements during training. Our setup is illustrated in figure 1.

GPM partitions each layer's gradient space into two orthogonal subspaces: Core Gradient Space (CGS) and Residual Gradient Space (RGS) as shown in 1(c). Important gradient directions (CGS) for previous tasks

Table 1: Comparison with prior works in Continual Learning (CL), Federated Learning (FL) and Decentralized Learning (DL) paradigm. Note that FL techniques require a server, while DL techniques are serverless. (†) denotes decentralized agents independently learning different tasks at a given time, and communicating via a fully-connected topology.

| Technique | CL | FL | DL |
|---|:---:|:---:|:---:|
| FedOpt (Konečný et al., 2016) | | ✓ | |
| DPSGD (Lian et al., 2017) | | | ✓ |
| EWC (Kirkpatrick et al., 2017) | ✓ | | |
| GPM (Saha et al., 2021) | ✓ | | |
| FedWeIT (Yoon et al., 2021) | ✓ | ✓ | |
| FLwF-2T (Usmanova et al., 2021) | ✓ | ✓ | |
| DCIL (Zhang et al., 2022) | ✓ | ✓ | |
| SKILL (Ge et al., 2023) | ✓ | | † |
| **CoDeC(ours)** | ✓ | | ✓ |

are stored in gradient projection memory (GPM), and gradient updates for the new tasks are taken along RGS to minimize interference. We find the basis vectors that span RGS and represent model updates as a linear combination of these vectors. We communicate the coefficients associated with these basis vectors instead of the model updates and achieve lossless communication compression. Further, theoretical insights into the convergence rate of CoDeC prove that it is possible to achieve similar rates as the state-of-the-art decentralized learning approaches such as DPSGD (Lian et al., 2017). We provide empirical evidence by performing experiments over various standard image-classification datasets, networks, graph sizes, and topologies. We also extend our analysis by designing and evaluating a decentralized continual learning benchmark MedMNIST-5 using biomedical image-classification datasets from MedMNIST-v2 (Yang et al., 2021). This imitates a practical real-world application where multiple healthcare organizations aim to learn a global generalized model without sharing the locally accessible patients' data. These models need to be updated with the emergence of variants of a disease, new diseases, or new diagnostic methods in a continual manner.

**Contributions:** We summarize our contributions as follows:

- We propose CoDeC, a communication-efficient decentralized continual learning algorithm that addresses a challenging problem: leveraging spatially and temporally distributed data to optimize a global model while preserving data privacy.
- We introduce a novel lossless communication compression scheme based on gradient subspaces.
- We theoretically show that our algorithm converges at the rate of $O(1/\sqrt{NK})$, where $N$ is the number of agents and $K$ is the number of training iterations.
- Experiments over a variety of image-classification datasets, networks, graph sizes, and topologies demonstrate minimal forgetting and up to $4.8\times$ reduction in communication costs with iso-performance relative to the full communication baseline.

## 2 Related Work

### 2.1 Decentralized Learning

Several works exist in the decentralized learning paradigm which enable distributing training without utilizing a central server (Bianchi et al., 2013; Lan et al., 2017; Lian et al., 2017; Assran et al., 2019; Balu et al., 2021). DPSGD (Lian et al., 2017) provides theoretical analysis for the convergence rate of decentralized learning algorithms, proving it to be similar to their centralized counterpart (Dean et al., 2012). In CoDeC, we utilize DPSGD (Lian et al., 2017) and modify it to send model updates instead of model parameters. Note, these existing works are not equipped to learn a temporal task sequence without forgetting past knowledge.

To reduce the communication overhead for decentralized learning, several compression techniques (Koloskova et al., 2019; Tang et al., 2019; Aketi et al., 2021) have been explored. DeepSqueeze (Tang et al., 2019) introduced error-compensated communication compression to decentralized training. Choco-SGD (Koloskova et al., 2019) communicates compressed model updates rather than parameters and achieves better performance than Tang et al. (2019). However, it is orthogonal to the compression scheme we present and can be used in synergy with our approach. Moreover, all of the above-mentioned compression techniques are lossy and require additional hyperparameter tuning.

## 2.2 Continual Learning

The majority of continual learning works fall into three categories (Wickramasinghe et al., 2023): network expansion, replay and regularization-based methods. Network expansion based methods (Rusu et al., 2016; Lee et al., 2017) overcome catastrophic forgetting by dedicating different model parameters to each task. Replay-based methods store training samples from the past tasks in the memory or synthesize old data from generative models for rehearsal (Chaudhry et al., 2019; Rebuffi et al., 2017; Shin et al., 2017). Regularization-based methods penalize changes to parameters  (Kirkpatrick et al., 2017; Zenke et al., 2017), or constrain gradient directions  (Saha et al., 2021; Wang et al., 2021; Saha & Roy, 2023) important for previous tasks. These methods rely on the availability of centrally located training data and hence fail to be directly applicable to a distributed learning scenario. Network expansion based methods in a decentralized continual learning setup may give rise to model heterogeneity across agents over time, while replay-based methods can lead to privacy concerns. Thus, we explore regularization based methods like GPM (Saha et al., 2021), SGP (Saha & Roy, 2023), EWC (Kirkpatrick et al., 2017) and SI (Zenke et al., 2017) in this work. We utilize GPM in CoDeC and show superior performance than D-EWC and D-SI, decentralized continual learning baselines we implemented with EWC and SI respectively. We further extend CoDeC to incorporate scaled gradient updates as shown in SGP.

## 2.3 Distributed Continual Learning

FedWeIT (Yoon et al., 2021) tackled the problem of federated continual learning through the decomposition of model parameters at each client into global and local task-adaptive parameters. FLwF-2T (Usmanova et al., 2021) developed a distillation-based method for class-incremental federated continual learning. Unlike our serverless training setup, these works utilize a central server to aggregate and send global updates to the agents. CoLLA (Rostami et al., 2017) focused on multi-agent distributed lifelong learning and proposed a distributed optimization algorithm for a network of synchronous learning agents. Note that it uses parametric models and is not directly applicable to modern deep neural networks. SKILL (Ge et al., 2023) proposes a distributed lifelong learning mechanism, where each agent uses a common pre-trained backbone and learns a task-specific head module. After training, these task-specific heads are shared among agents via a fully connected graph. However, each SKILL agent is independently learning a different task at a given time, lacking the concept of collaborative learning as demonstrated in CoDeC.

# 3 Methodology

## 3.1 Problem Formulation

In this work, we optimize a DNN model to learn from spatially and temporally distributed data. The communication topology is modeled as a graph $G = ([N], \mathbf{W})$, where $N$ is the number of learning agents and $\mathbf{W}$ is the mixing matrix indicating the graph's connectivity. $w_{ij}$ encodes the effect of agent $j$ on agent $i$, and $w_{ij} = 0$ implies there is no direct communication link between the two agents.

Consider a learning scenario where $T$ tasks are learned sequentially. Now, for any task $\tau \in \{1, .., T\}$, the corresponding dataset $\mathcal{D}_\tau$ is independently and identically distributed (IID) across the $N$ agents as $\{\mathcal{D}_{\tau,1}, \mathcal{D}_{\tau,2}, \mathcal{D}_{\tau,3}.....\mathcal{D}_{\tau,N}\}$. For each $\tau$, we aim to minimize the global loss function $\mathcal{F}_\tau(\mathbf{x})$ given in equation 1. Here, $F_{\tau,i}(d_{\tau,i}, \mathbf{x})$ is the local loss function per task at agent $i$ and $f_{\tau,i}(\mathbf{x})$ is the expected value of $F_{\tau,i}(d_{\tau,i}, \mathbf{x})$ over the dataset $\mathcal{D}_{\tau,i}$.

$$\min_{\mathbf{x}\in\mathbb{R}^d} \mathcal{F}_\tau(\mathbf{x}) = \frac{1}{N}\sum_{i=1}^N f_{\tau,i}(\mathbf{x}),$$

$$where \quad f_{\tau,i}(\mathbf{x}) = \mathbb{E}_{d_{\tau,i}\sim\mathcal{D}_{\tau,i}}[F_{\tau,i}(d_{\tau,i},\mathbf{x})] \ \ \forall i \tag{1}$$

Decentralized optimization of this global loss function $\mathcal{F}_\tau(\mathbf{x})$ is based on the current dataset $\mathcal{D}_\tau$. A crucial challenge is to optimize $\mathcal{F}_\tau(\mathbf{x})$ such that the past information acquired from tasks $1, 2, .., (\tau - 1)$ is retained. Inspired by Saha et al. (2021), we define a subspace that contains important gradient directions associated with all the past tasks and modify the local gradient updates of the current task to be orthogonal to this subspace i.e., to lie in RGS. This ensures minimal catastrophic forgetting.

Typically, decentralized agents communicate the model parameters with their neighbors in each training iteration (Lian et al., 2017). Note that in the proposed algorithm the model updates lie in RGS, which is a smaller vector subspace compared to the entire gradient space. To utilize this property for enabling lossless communication compression, we communicate model updates with neighbors similar to Koloskova et al. (2019) rather than the model parameters.

### 3.2 Approach

We demonstrate the flow of CoDeC in Algorithm 1. All hyperparameters are synchronized between the agents at the beginning of the training. Each agent $i$ computes the gradient update $\mathbf{g}^i = (\nabla f_{\tau,i}(d_{\tau,i}; \mathbf{x}^i))$ with respect to model parameters $\mathbf{x}^i$, evaluated on mini-batch $d_{\tau,i}$. We obtain $\tilde{\mathbf{g}}^i$, the orthogonal projection of the local gradients using GPM memory $\mathcal{M}$ (line 6, algorithm 1). The parameters of each agent are updated using this $\tilde{\mathbf{g}}^i$ which ensures minimal forgetting. Then, each agent performs a gossip averaging step using $\mathbf{x}^i$ and $\hat{\mathbf{x}}^j$ (line 8, algorithm 1). $\hat{\mathbf{x}}^j$ represent the copies of $\mathbf{x}^j$ maintained by all the neighbors of agent $j$ and in general $\mathbf{x}^j = \hat{\mathbf{x}}^j$. The computed model updates (denoted by $\mathbf{q}_i^k$) lie in the RGS subspace spanned by the basis vectors contained in $\mathbf{O}^l$. Therefore, we express them as a linear combination of these basis vectors and find the associated coefficients, $\mathbf{c}^i$ to communicate with the neighbors (line 10, algorithm 1). Upon receiving these coefficients, the agents reconstruct the neighbors' updates without any loss in information (line 13, algorithm 1). Communicating the coefficients ($\mathbf{c}^i$) leads to lossless compression, which we elaborate upon in section 3.3. The local copy $\hat{\mathbf{x}}^j$ is updated using the reconstructed model updates $\mathbf{q}^j$ (line 14, algorithm 1). Note that our algorithm requires each agent to only store the sum of neighbors' models $\sum_{j\in\mathcal{N}(i)} w_{ij}\hat{\mathbf{x}}^j$ resulting in $O(1)$ memory overhead, independent of the number of neighbors.

At the end of each task, important gradient directions are obtained using a Singular Value Decomposition (SVD) representation of the input activations of each layer (Saha et al., 2021). These gradient directions are added as basis vectors to the CGS matrix $\mathcal{M}$ and subsequently removed from the RGS Matrix $\mathcal{O}$. SVD is calculated using a subset of training data at any randomly chosen agent and communicated to other agents iteratively using the communication graph.

### 3.3 Lossless Compression

Stochastic Gradient Descent (SGD) updates lie in the span of input data points (Zhang et al., 2017). Leveraging this fact, GPM (Saha et al., 2021) performs SVD on a representation matrix $\mathbf{R}_\tau^l$ and finds basis vectors corresponding to the important gradient directions for previous tasks. $\mathbf{R}_\tau^l$ is constructed by performing a forward pass of $n_s$ samples from the training dataset for task $\tau$ through the network and concatenating the input activations for each layer $l$ (equation 2). Subsequently, the SVD of representation, $\mathbf{R}_\tau^l$ in equation 2 is used to obtain the matrix $\mathbf{U}_\tau^l$ containing a set of orthonormal basis vectors which span the entire gradient space.

$$\mathbf{R}_\tau^l = [x_{1,\tau}^l, x_{2,\tau}^l.., x_{n_s,\tau}^l]$$

$$SVD(\mathbf{R}_\tau^l) = \mathbf{U}_\tau^l \mathbf{\Sigma}(\mathbf{V}_\tau^l)^T \tag{2}$$

The threshold hyperparameter $\epsilon_{th}$ determines the number of basis vectors chosen from $\mathbf{U}_\tau^l$ to represent important gradient directions for any particular task. These vectors span a subspace in the gradient space which we define as the Core Gradient Space (CGS). They are added to the GPM matrix $\mathcal{M} = \{(\mathbf{M}^l)_{l=1}^L\}$.

---

**Algorithm 1** Communication-Efficient Decentralized Continual Learning (*CoDeC*)

---

**Input:** Each agent $i \in [1, N]$ initializes model parameters $\mathbf{x}_0^i$, step size $\eta$, mixing matrix $\mathbf{W} = [w_{ij}]_{i,j \in [1,N]}$, $\hat{\mathbf{x}}_{(0)}^i = \mathbf{x}_0^i$, $\mathbf{M}^l = [\ ]$ and $\mathbf{O}^l = [\mathbf{I}]$ for all layers $l = 1, 2, ...L$, GPM Memory $\mathcal{M} = \{(\mathbf{M}^l)_{l=1}^L\}$, RGS Matrix $\mathcal{O} = \{(\mathbf{O}^l)_{l=1}^L\}$, $\mathcal{N}(i)$: neighbors of agent $i$ (including itself), $T$: total tasks, $K$: training iterations

Each agent simultaneously implements the TRAIN( ) procedure
1. **procedure** TRAIN( )
2.     **for** $\tau = 1, \ldots, T$ **do**
3.         **for** $k = 0, 1, \ldots, K - 1$ **do**
4.             $d_{\tau,i} \sim \mathcal{D}_{\tau,i}$
5.             $\mathbf{g}_k^i = \nabla f_{\tau,i}(d_{\tau,i}; \mathbf{x}_k^i)$
6.             $\tilde{\mathbf{g}}_k^i = \mathbf{g}_k^i - (\mathbf{M}^l(\mathbf{M}^l)^T)\mathbf{g}_k^i$  # for each layer $l$
7.             $\mathbf{x}_{(k+\frac{1}{2})}^i = \mathbf{x}_k^i - \eta \tilde{\mathbf{g}}_k^i$
8.             $\mathbf{x}_{k+1}^i = \mathbf{x}_{(k+\frac{1}{2})}^i + \sum_{j \in \mathcal{N}(i)} w_{ij}(\hat{\mathbf{x}}_k^j - \mathbf{x}_k^i)$
9.             $\mathbf{q}_k^i = \mathbf{x}_{k+1}^i - \mathbf{x}_k^i$
10.           $\mathbf{c}_k^i = (\mathbf{O}^l)^T \mathbf{q}_k^i$
11.           for each $j \in \mathcal{N}(i)$ **do**
12.              Send $\mathbf{c}_k^i$ and receive $\mathbf{c}_k^j$
13.              $\mathbf{q}_k^j = (\mathbf{O}^l)\mathbf{c}_k^j$
14.              $\hat{\mathbf{x}}_{(k+1)}^j = \mathbf{q}_k^j + \hat{\mathbf{x}}_k^j$
15.           **end**
16.         **end**
          # GPM Update
17.         $p = random(1, 2, ...N)$
18.         **if** $i == p$ **do**
19.           Update $\mathbf{M}^l$, $\mathbf{O}^l$ for each layer $l \in L$
20.           Update $\mathcal{M} = \{(\mathbf{M}^l)_{l=1}^L\}$
21.           Update $\mathcal{O} = \{(\mathbf{O}^l)_{l=1}^L\}$
22.           Send $\mathcal{M}, \mathcal{O}$ to all agents
23.         **end**
24.     **end**
25. **return**

---

For task $\tau > 1$, we ensure that the CGS vectors being added to the GPM matrix in round $\tau$ are orthogonal to all the CGS vectors in stored in $\mathbf{M}$. Before performing SVD on the representation matrix $\mathbf{R}_\tau^l$ for each layer $l$, we perform the following projection step:

$$\hat{\mathbf{R}}_\tau^l = \mathbf{R}_\tau^l - (\mathbf{M}^l(\mathbf{M}^l)^T)\mathbf{R}_\tau^l \tag{3}$$

SVD is then performed on $\hat{\mathbf{R}}_\tau^l$ and new orthogonal basis vectors are added to $\mathbf{M}$. This ensures that the newly added basis vectors for task $\tau$ are unique and orthogonal to the vectors already present in $\mathbf{M}$.

The following update rule is used to obtain orthogonal gradient update $\tilde{\mathbf{g}}^i$ for the later tasks:

$$\tilde{\mathbf{g}}^i = \mathbf{g}^i - (\mathbf{M}^l(\mathbf{M}^l)^T)\mathbf{g}^i \tag{4}$$

Here, $\mathbf{g}^i$ is the original local gradient update at agent $i$ at layer $l$, and the projection of $\mathbf{g}^i$ on CGS is $(\mathbf{M}^l \mathbf{M}^{l^T})\mathbf{g}^i$. Let the input space for a layer be of dimension $n_l$. This implies that $\mathbf{U}_\tau^l$ contains $n_l$ orthonormal basis vectors. Now based on $\epsilon_{th}$, after every task, a set of $r_l$ basis vectors corresponding to the top $r_l$ singular values are stored in $\mathcal{M}$. Hence, $\tilde{\mathbf{g}}^i$ lies in a $(n_l - r_l)$ dimensional orthogonal subspace denoted as the Residual Gradient Space (RGS).

The basis vectors which span RGS are the remaining $(n_l - r_l)$ vectors contained in $\mathbf{U}_\tau^l$. We store them in the RGS Matrix $\mathcal{O} = \{(\mathbf{O}^l)_{l=1}^L\}$. $n_l - r_l < n_l$, and $r_l$ increases as the task sequence progresses. We note that the gradient updates tend to lie in a smaller subspace (i.e. RGS) whose dimensionality decreases based on $\epsilon_{th}$ and the number of tasks.

In algorithm 1, model updates $\mathbf{q}_k^i$ are computed at every training iteration $k$. Since all the local gradients $\tilde{\mathbf{g}}_k^i$ lie in RGS, the updates $\mathbf{q}_k^i$ also lie in RGS (refer to A.6 for the proof). Therefore, we express layer-wise $\mathbf{q}_k^i$ as a linear combination of the basis vectors in $\mathbf{O}^l$ and find the associated coefficients $\mathbf{c}_k^i$. The neighbors of agent $i$ reconstruct the updates $\mathbf{q}_k^i$ from the received $\mathbf{c}_k^i$. This encoding and decoding of $\mathbf{q}_k^i$ require two additional matrix multiplications (lines 10 and 13 in algorithm 1). Our approach ensures that all agents have the same $\mathcal{M}$ and $\mathcal{O}$ matrices so that the reconstruction is exact. Hence, we achieve lossless communication compression by the virtue of taking orthogonal gradient updates to avoid catastrophic forgetting.

## 4 Convergence Rate Analysis

In this section, we provide a convergence analysis for our algorithm. In particular, we provide an upper bound for $\|\nabla\mathcal{F}(\bar{\mathbf{x}}_k)\|^2$, where $\nabla\mathcal{F}(\bar{\mathbf{x}}_k)$ is the average gradient achieved by the averaged model across all agents. Since our claims are valid for each task, the task subscript is dropped for the following analysis. We make the following assumptions:

**Assumption 1 - Lipschitz Gradients:** Each function $f_i(\mathbf{x})$ is L-smooth.

**Assumption 2 - Bounded Variance:** The variance of the stochastic gradients is assumed to be bounded. There exist constants $\sigma$ and $\delta$ such that

$$\mathbb{E}_{d\sim\mathcal{D}_i}||\nabla F_i(\mathbf{x};d) - \nabla f_i(\mathbf{x})||^2 \le \sigma^2 \tag{5}$$

$$\frac{1}{N}\sum_{i=1}^{N}||\nabla f_i(\mathbf{x}) - \nabla\mathcal{F}(\mathbf{x})||^2 \le \delta^2 \ \ \forall i, x \tag{6}$$

**Assumption 3 - Doubly Stochastic Mixing Matrix:** $\mathbf{W}$ is a real doubly stochastic matrix with $\lambda_1(\mathbf{W}) = 1$ and $max\{|\lambda_2(\mathbf{W})|, |\lambda_N(\mathbf{W})|\} \le \sqrt{\rho} < 1$, where $\lambda_i(\mathbf{W})$ is the $i^{th}$ largest eigenvalue of $\mathbf{W}$ and $\rho$ is a constant.

The above assumptions are commonly used in most decentralized learning works (Lian et al., 2017; Tang et al., 2019; Esfandiari et al., 2021). Since we modify the original gradient update $\mathbf{g}^i$, we introduce an additional assumption:

**Assumption 4 - Bounded Orthogonal Updates:** For each agent $i$, we have:

$$\|\tilde{\mathbf{g}}^i\| \le \mu\|\mathbf{g}^i\| \tag{7}$$

where $\mu \in (0, 1]$ signifies how constrained the gradient space is. In particular, $\mu$ encapsulates the average impact of the dimension of RGS subspace during training.

To ensure that the gradient update after projection is in the descent direction, we provide the following lemma:

**Lemma 1.** *Given the original gradient update* -$\mathbf{g}^i$ *is in the descent direction, the orthogonal gradient update* -$\tilde{\mathbf{g}}^i$ *is also in the descent direction.*

Please refer to Appendix A.1 for the proof. Theorem 2 presents the convergence of CoDeC (proof in Appendix A.3).

**Theorem 2.** *Given assumptions 1-4, let step size $\eta$ satisfy the following condition:*

$$\frac{1}{L} < \eta \le \frac{\sqrt{(1-\sqrt{\rho})^2 + 12\mu^2} - (1-\sqrt{\rho})}{6L\mu^2} \tag{8}$$

*For all $K \ge 1$, we have*

$$\frac{1}{K}\sum_{k=0}^{K-1}\mathbb{E}\left[\|\nabla\mathcal{F}(\bar{\mathbf{x}}_k)\|^2\right] \le \frac{1}{C_1 K}\left(\mathbb{E}\left[\mathcal{F}(\bar{\mathbf{x}}_0) - \mathcal{F}^*\right]\right) +$$
$$C_2\frac{\sigma^2}{N} + C_3\,\eta^2\mu^2\left(\frac{3\sigma^2}{(1-\sqrt{\rho})^2} + \frac{3\delta^2}{(1-\sqrt{\rho})^2}\right) \tag{9}$$

*where $C_1 = \frac{1}{2}(\eta - \frac{1}{L})$, $C_2 = L\eta^2/2C_1$, $C_3 = L^2\eta/2C_1$.*

The result of theorem 2 shows that the norm of the average gradient achieved by the consensus model is upper-bounded by the suboptimality gap $(\mathcal{F}(\bar{\mathbf{x}}_0) - \mathcal{F}^*)$, the sampling variance $(\sigma)$, the gradient variations $(\delta)$, and the constraint on the gradient space $(\mu)$. The suboptimality gap signifies how good the model initialization is. $\sigma$ indicates the variation in gradients due to stochasticity, while $\delta$ is related to gradient variations across the agents. From equation 9, we observe that $\mu$ appears in the last term and effectively scales $\sigma$ and $\delta$. A detailed explanation of the constraints on step size $\eta$ is presented in Appendix A.4. We present a corollary to show the convergence rate of CoDeC in terms of the training iterations. Note that we denote $a_n = O(b_n)$ if $a_n \leq c b_n$, where $c > 0$ is a constant.

**Corollary 3.** *Suppose that the step size satisfies $\eta = O\left(\sqrt{\frac{N}{K}}\right)$. For a sufficiently large $K$ and some constant $C > 0$,*

$$\frac{1}{K} \sum_{k=0}^{K-1} \mathbb{E}\left[\|\nabla \mathcal{F}(\bar{\mathbf{x}}_k)\|^2\right] \leq C\left(\frac{1}{\sqrt{NK}} + \frac{1}{K}\right) \tag{10}$$

The proof for Corollary 3 is detailed in Appendix A.5. It indicates that CoDeC achieves a convergence rate of $O(\frac{1}{\sqrt{NK}})$ for each task. This rate is similar to the well-known best result in decentralized SGD algorithms (Lian et al., 2017). Since $\mu^2$ appears only in the higher order term $\frac{1}{K}$, it does not affect the order of the convergence rate.

## 5 Experimental Setup

**Implementation details:** For each task, the data distribution is IID across the agents. The agents communicate with their neighbors after every mini-batch update. We present results for different graph topologies and sizes: directed ring with 4/8/16 agents and undirected torus with 8/16 agents. In the directed ring topology, each agent has only 1 neighbor. Meanwhile, the torus topology has higher connectivity, with 3 and 4 neighbors for graph sizes of 8 and 16 agents respectively. We evaluate CoDeC on three well-known continual learning benchmark datasets: 10-Split CIFAR-100 (Krizhevsky, 2009), 20-Split MiniImageNet (Vinyals et al., 2016) and a sequence of 5-Datasets (Ebrahimi et al., 2020). 10-Split CIFAR-100 is constructed by splitting CIFAR-100 into 10 tasks, where each task comprises of 10 classes. We use a 5-layer AlexNet for experiments with Split CIFAR-100. 20-Split miniImageNet has 20 sequential tasks, where each task comprises 5 classes. The sequence of 5-Datasets includes CIFAR-10, MNIST, SVHN (Netzer et al., 2011), notMNIST (Bulatov, 2011) and Fashion MNIST (Xiao et al., 2017). For Split miniImageNet and 5-Datasets, we use a reduced ResNet18 architecture similar to Lopez-Paz & Ranzato (2017). We design MedMNIST-5, a biomedical decentralized continual learning benchmark based on the datasets in MedMNIST-v2 (Yang et al., 2021). MedMNIST-5 consists of the following sequential classification tasks: TissueMNIST, OrganAMNIST, OCTMNIST, PathMNIST, BloodMNIST. ResNet-18 architecture is used to evaluate the performance. For all experiments, batch normalization parameters are learned for the first task and frozen for subsequent tasks. We use 'multi-head' setting, where each task has a separate final classifier with no constraints on gradient updates. Please refer to Appendix A.7, A.8, A.9 for details related to architectures, dataset statistics, and training hyperparameters, respectively.

**Baselines:** To the best of our knowledge, our proposed setup is unique and hence there are no directly comparable baselines. Therefore, for a fair comparison we implement two baselines D-EWC and D-SI based on well known continual learning techniques Elastic Weight Consolidation (EWC) (Kirkpatrick et al., 2017) and Synaptic Intelligence (SI) (Zenke et al., 2017) respectively. Please refer to algorithm 2 and 3 for the detailed implementation. At the end of each task, these methods compute statistics to penalize the parameter updates to mitigate forgetting. D-EWC computes Fisher information matrix $\mathcal{F}$, while D-SI utilizes parameter specific contribution to changes in the total loss to compute importance measure $\Omega$. To provide an upper bound on performance, we add two baselines: STL and D-STL. Single Task Learning (STL) represents a setting where all the tasks are learned sequentially in a centralized setup without any constraints. Decentralized Single Task Learning (D-STL) baseline extends STL to a decentralized setting.

**Performance Metrics:** We mainly focus on these metrics: (1) *Average Accuracy (ACC)*: measures the average test classification accuracy of all tasks (2) *Backward Transfer (BWT)*: indicates the impact on

the past knowledge after learning new tasks where negative BWT implies forgetting (3) *Communication Compression (CC)*: measures the relative reduction in the communication cost achieved through our lossless compression scheme. ACC and BWT can be formally defined as:

$$\text{ACC} = \frac{1}{T}\sum_{i=1}^{T} A_{T,i}; \text{BWT} = \frac{1}{T-1}\sum_{i=1}^{T-1} A_{T,i} - A_{i,i} \tag{11}$$

Here, T is the total number of tasks and $A_{T,i}$ is the accuracy of the model on $i^{th}$ task after learning T tasks sequentially.

## 6 Results and Discussions

**Continual Learning Statistics Aggregation:** We implement two versions for each technique based on how continual learning statistics are computed and consolidated at the end of each task: broadcast and all-gather. In broadcast, an agent is randomly chosen and the corresponding statistics are calculated and sent to all the other agents. In all-gather, all agents compute these statistics using their local data, and the global average of these statistics is utilized to mitigate forgetting. Note that all-gather incurs more computational cost as compared to broadcast. D-SI and D-EWC give a sub-optimal performance with broadcast, while CoDeC gives a similar performance for both versions as shown in Table 2 (refer to Table 11 in Appendix for additional results). Hence, for results in Table 3, 4, 5 and 6 we choose all-gather for D-EWC and D-SI, and broadcast for CoDeC.

Table 2: Impact of broadcast and all-gather for continual learning for Split miniImageNet over 8 agent directed ring.

| Setup | ACC(%) | BWT(%) |
|---|---|---|
| D-SI (broadcast) | 36.14 ± 1.61 | -12.90 ± 1.15 |
| D-SI (all-gather) | 45.58 ± 1.24 | -3.67 ± 1.27 |
| D-EWC (broadcast) | 37.20 ± 0.58 | 0.27 ± 0.15 |
| D-EWC (all-gather) | 46.39 ± 1.54 | -1.64 ± 1.11 |
| CoDeC(broadcast) | 53.22 ± 1.82 | 0.08 ± 0.45 |
| CoDeC (all-gather) | 53.25 ± 1.48 | 0.50 ± 0.21 |

**Performance Comparison:** We present two versions of our approach: CoDeC, which uses the lossless compression scheme, and CoDeC(f), an implementation with full communication. As shown in Table 3, for Split CIFAR-100 we obtain 3-4% better ACC than D-EWC with a similar order of BWT. CoDeC achieves 12-13% better ACC than D-SI, with marginally better BWT. Our proposed compression technique results in a 1.86x reduction in the communication cost on average without any performance degradation.

Table 4 demonstrates learning a longer task sequence Split MiniImageNet, and we outperform both D-EWC and D-SI by 6-11% in terms of ACC with better BWT in some cases. We achieve 1.43x reduction in communication cost on average over a range of graph sizes. Learning dynamics of task 1 after learning each task in Split miniImageNet are shown in figure 2. Note that for D-SI, the training converges only for a lower learning rate as compared to D-EWC and CoDeC (details in Appendix A.9). Results on 5-Datasets demonstrate learning across diverse datasets. As shown in Table 5, although we report better BWT for D-EWC and D-SI in most cases, we achieve 0.5-5% better accuracy and upto 2.2x reduced communication cost. We believe that final average accuracy (ACC) and backward transfer (BWT) are both equally important in continual learning scenarios. It is possible to achieve zero BWT by freezing the model weights after learning the first task. However, that leads to sub-optimal ACC due to lack of ability to learn the subsequent tasks effectively. In our setup, we strive to achieve a balance between these two metrics by tuning the threshold hyperparameter $\epsilon_{th}$ accordingly. It is possible to use a higher $\epsilon_{th}$ and achieve a lower BWT, but at the cost of lower ACC. Hence, $\epsilon_{th}$ can be tuned to achieve lower BWT or higher ACC as required.

Additionally, we present results on MedMNIST-5, a decentralized continual learning benchmark we propose to imitate a scenario where healthcare organizations aim to optimize a global model while maintaining

Table 3: Split CIFAR-100 over AlexNet using directed ring and torus. ($^*$): methods that don't adhere to CL setup and provide an upper-bound on the performance.

| Agents | Setup | Directed Ring | | | Torus | | |
|---|---|---|---|---|---|---|---|
| | | ACC(%) | BWT(%) | CC | ACC(%) | BWT(%) | CC |
| | STL$^*$ | 70.56 ± 0.20 | - | - | - | - | - |
| 4 | D-STL$^*$ | 69.22 ± 0.10 | - | - | - | - | - |
| | D-SI | 44.90 ± 0.10 | -0.83 ± 0.64 | 1x | - | - | - |
| | D-EWC | 53.12 ± 0.62 | 0.24 ± 0.18 | 1x | - | - | - |
| | CoDeC(f) | 57.54 ± 0.25 | -1.22 ± 0.22 | 1x | - | - | - |
| | CoDeC | 57.83 ± 0.25 | -0.95 ± 0.05 | 1.86x | - | - | - |
| 8 | D-STL$^*$ | 64.99 ± 0.41 | - | - | 65.17 ± 0.44 | - | - |
| | D-SI | 39.54 ± 0.16 | -1.08 ± 0.85 | 1x | 39.36 ± 0.40 | -1.25 ± 0.65 | - |
| | D-EWC | 50.52 ± 0.58 | 0.51 ± 0.09 | 1x | 49.41 ± 0.88 | 0.29 ± 0.27 | 1x |
| | CoDeC(f) | 53.57 ± 0.38 | -0.65 ± 0.52 | 1x | 53.54 ± 0.35 | -1.15 ± 0.41 | 1x |
| | CoDeC | 53.63 ± 0.25 | -0.43 ± 0.33 | 1.85x | 53.62 ± 0.29 | -0.64 ± 0.36 | 1.86x |
| 16 | D-STL$^*$ | 58.31 ± 0.49 | - | - | 59.29 ± 0.12 | - | - |
| | D-SI | 34.66 ± 1.15 | -1.23 ± 0.4 | 1x | 34.86 ± 0.68 | -1.16 ± 0.54 | 1x |
| | D-EWC | 45.52 ± 0.60 | 0.22 ± 0.34 | 1x | 44.53 ± 0.77 | -0.20 ± 0.56 | 1x |
| | CoDeC(f) | 48.05 ± 0.45 | -0.38 ± 0.12 | 1x | 48.19 ± 0.27 | -0.29 ± 0.11 | 1x |
| | CoDeC | 48.16 ± 0.33 | -0.18 ± 0.28 | 1.84x | 48.36 ± 0.04 | -0.26 ± 0.31 | 1.84x |

Table 4: Split MiniImagenet over ResNet-18 using directed ring and torus. ($^*$): methods that don't adhere to CL setup.

| Agents | Setup | Directed Ring | | | Torus | | |
|---|---|---|---|---|---|---|---|
| | | ACC(%) | BWT(%) | CC | ACC(%) | BWT(%) | CC |
| | STL$^*$ | 70.18 ± 2.75 | - | - | | | |
| 4 | D-STL$^*$ | 69.36 ± 0.78 | - | - | - | - | - |
| | D-SI | 51.01 ± 0.86 | -4.00 ± 0.61 | 1x | - | - | - |
| | D-EWC | 52.81 ± 2.80 | -1.07 ± 2.03 | 1x | - | - | - |
| | CoDeC(f) | 60.03 ± 0.75 | 0.36 ± 1.01 | 1x | - | - | - |
| | CoDeC | 59.00 ± 2.56 | -0.79 ± 0.27 | 1.51x | - | - | - |
| 8 | D-STL$^*$ | 63.13 ± 0.86 | - | - | 66.27 ± 1.47 | - | - |
| | D-SI | 45.58 ± 1.24 | -3.67 ± 1.27 | 1x | 46.00 ± 0.73 | -3.21 ± 0.51 | 1x |
| | D-EWC | 46.39 ± 1.54 | -1.64 ± 1.11 | 1x | 48.23 ± 3.14 | -1.02 ± 1.16 | 1x |
| | CoDeC(f) | 53.22 ± 1.82 | 0.08 ± 0.45 | 1x | 59.90 ± 0.48 | 0.37 ± 0.24 | 1x |
| | CoDeC | 53.30 ± 1.25 | -0.46 ± 0.48 | 1.37x | 59.97 ± 0.87 | -0.19 ± 0.98 | 1.53x |
| 16 | D-STL$^*$ | 57.09 ± 1.55 | - | - | 63.51 ± 0.61 | - | - |
| | D-SI | 39.55 ± 0.87 | -2.03 ± 0.69 | 1x | 39.96 ± 0.47 | -1.74 ± 0.96 | 1x |
| | D-EWC | 39.67 ± 1.37 | -1.32 ± 1.18 | 1x | 45.14 ± 0.18 | -0.64 ± 0.23 | 1x |
| | CoDeC(f) | 45.29 ± 3.58 | -0.99 ± 1.40 | 1x | 51.03 ± 2.51 | -0.01 ± 0.67 | 1x |
| | CoDeC | 45.68 ± 0.77 | 0.61 ± 0.79 | 1.42x | 51.32 ± 1.05 | 0.26 ± 0.56 | 1.39x |

patient's privacy. Results in Table 6 show that CoDeC outperforms D-EWC and D-SI by 7% and 1% respectively in terms of ACC while incurring minimum BWT. The lossless compression scheme results in a 2x reduction in the communication cost. In all our experiments, we observe that ACC decreases as we increase the graph size, while BWT remains of the similar order.

We also compare CoDeC with SKILL, a distributed lifelong learning algorithm (results in Table 12). Furthermore, we present results on non-IID data distribution across the agents in Table 13.

**Task-wise Communication Costs**: The reduction in communication cost is a reflection of the constraints on the direction of gradient updates. As the gradient updates are not constrained for the first task, they occupy the entire gradient space. However, gradient updates after learning task 1 are constrained to the RGS subspace, whose dimensionality decreases as the task sequence progresses. This implies an increase

Table 5: 5-Datasets over ResNet-18 using directed ring and torus. (*): methods that don't adhere to CL setup.

| Agents | Setup | Directed Ring | | | Torus | | |
|---|---|---|---|---|---|---|---|
| | | ACC(%) | BWT(%) | CC | ACC(%) | BWT(%) | CC |
| | STL* | 92.79 ± 0.08 | - | - | - | - | - |
| 4 | D-STL* | 92.51 ± 0.18 | - | - | - | - | - |
| | D-SI | 82.44 ± 0.29 | -1.52 ± 0.13 | 1x | - | - | - |
| | D-EWC | 86.82 ± 0.25 | -3.37 ± 0.80 | 1x | - | - | - |
| | CoDeC(f) | 87.24 ± 0.23 | -4.05 ± 0.05 | 1x | - | - | - |
| | CoDeC | 87.41 ± 0.44 | -4.03 ± 0.30 | 2.13x | - | - | - |
| 8 | D-STL* | 92.31 ± 0.06 | - | - | 92.32 ± 0.15 | - | - |
| | D-SI | 80.36 ± 0.15 | -2.64 ± 0.07 | 1x | 79.55 ± 0.33 | -3.07 ± 0.13 | 1x |
| | D-EWC | 85.69 ± 0.19 | -0.92 ± 0.14 | 1x | 82.99 ± 3.25 | -2.10 ± 1.60 | 1x |
| | CoDeC(f) | 86.54 ± 0.04 | -4.37 ± 0.17 | 1x | 85.92 ± 0.18 | -5.10 ± 0.17 | 1x |
| | CoDeC | 86.23 ± 0.22 | -4.61 ± 0.32 | 2.17x | 86.15 ± 0.17 | -4.85 ± 0.26 | 2.19x |
| 16 | D-STL* | 92.16 ± 0.16 | - | - | 91.76 ± 0.09 | - | - |
| | D-SI | 78.53 ± 0.62 | -5.2 ± 0.56 | 1x | 77.00 ± 0.11 | -5.76 ± 0.22 | 1x |
| | D-EWC | 82.19 ± 0.45 | -0.18 ± 0.05 | 1x | 81.48 ± 0.12 | -0.56 ± 0.14 | 1x |
| | CoDeC(f) | 86.36 ± 0.15 | -4.36 ± 0.19 | 1x | 84.91 ± 0.20 | -5.48 ± 0.22 | 1x |
| | CoDeC | 86.41 ± 0.16 | -4.37 ± 0.24 | 2.16x | 85.00 ± 0.55 | -5.52 ± 0.35 | 2.23x |

Table 6: MedMNIST-5 over ResNet-18 using 16 agent directed ring topology.

| Setup | ACC(%) | BWT(%) | CC |
|---|---|---|---|
| D-STL* | 76.55 ± 0.70 | - | - |
| D-SI | 61.53 ± 0.49 | -7.51 ± 0.72 | 1x |
| D-EWC | 55.62 ± 1.23 | -0.72 ± 0.25 | 1x |
| CoDeC(f) | 62.51 ± 0.46 | -0.37 ± 0.27 | 1x |
| CoDeC | 62.47 ± 0.13 | -0.07 ± 0.44 | 2.03x |

in compression ratios, which is clearly reflected in our results highlighting task-wise CC in figure 3. In essence, as the gradient space becomes more constrained, it suffices for agents to communicate less with their neighbors. Hence, we achieve a CC of 2.1x for task 2, with this increasing up to 4.8x for task 5.

**Scope of ACC-CC Trade-off:** It is possible to get better ACC with CoDeC, albeit at the cost of reduced CC. This is achieved by relaxing the orthogonal gradient constraint and allowing gradient updates along some CGS basis vectors, as demonstrated in Scaled Gradient Projection (SGP) (Saha & Roy, 2023). The gradient update along each CGS basis vector is scaled according to its importance $\lambda$ for the past tasks.

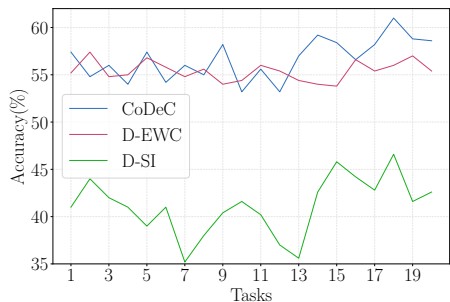

Figure 2: Evolution of task 1 accuracy over the course of 20 tasks from Split MiniImageNet over a 4 agent ring topology

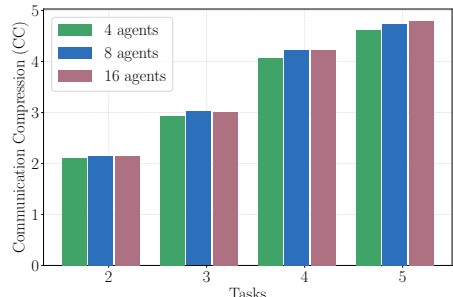

Figure 3: Task-wise CC for 5-Datasets over ResNet-18

Table 7: Split CIFAR-100 over AlexNet using directed ring and torus topologies with scaled gradient updates.

| Agents | Setup | Directed Ring | | | Torus | | |
|---|---|---|---|---|---|---|---|
| | | ACC(%) | BWT(%) | CC | ACC(%) | BWT(%) | CC |
| 4 | CoDeC(f) | 59.07 ± 0.51 | -2.19 ± 0.25 | 1x | - | - | - |
| | CoDeC | 59.20 ± 0.6 | -1.95 ± 0.26 | 1.42x | - | - | - |
| 8 | CoDeC(f) | 55.33 ± 0.21 | -1.02 ± 0.59 | 1x | 56.02 ± 0.4 | -0.88 ± 0.35 | 1x |
| | CoDeC | 55.68 ± 0.36 | -1.10 ± 0.27 | 1.42x | 55.69 ± 0.27 | -1.01 ± 0.06 | 1.42x |
| 16 | CoDeC(f) | 49.68 ± 0.51 | -0.51 ± 0.37 | 1x | 50.15 ± 0.12 | -0.09 ± 0.19 | 1x |
| | CoDeC | 49.56 ± 0.90 | -0.57 ± 0.30 | 1.42x | 50.00 ± 0.49 | -0.88 ± 0.23 | 1.42x |

The lower the $\lambda$, the lesser the importance of the corresponding basis direction. $\lambda=1$ implies no gradient steps can be taken along that basis direction. The following update rule is used to obtain the scaled gradient update $\tilde{\mathbf{g}}^i$:

$$\tilde{\mathbf{g}}^i = \mathbf{g}^i - (\mathbf{M}^l \Lambda (\mathbf{M}^l)^T)\mathbf{g}^i \qquad (12)$$

Here, $\Lambda$ is a diagonal matrix containing $\lambda$ in its diagonal. Note that $\Lambda = \mathbf{I}$ corresponds to the GPM update rule in equation 4. The computation of $\lambda$ involves a scale coefficient $\alpha$. $\lambda$ tends to 1 for large $\alpha$ values, and SGP converges to GPM. To incorporate gradient scaling in CoDeC, the basis vectors in $\mathcal{M}$

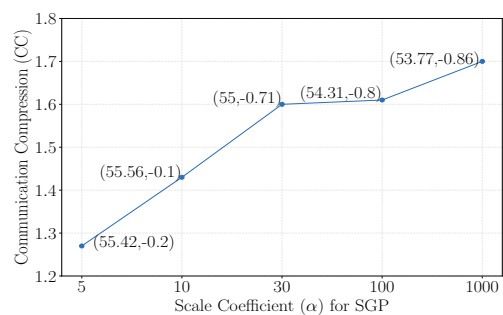

Figure 4: Impact of $\alpha$ on CC for Split CIFAR-100 over an 8 agent ring. The data labels denote (ACC, BWT).

with $\lambda < 1$ are appended to $\mathcal{O}$ at the end of each task. This provides us with a knob to achieve an ACC-CC trade-off, as shown in figure 4. We find that ACC increases as $\alpha$ decreases, albeit at the cost of lower CC. This is the result of the gradient updates lying in a larger subspace for smaller values of $\alpha$. For $\alpha=1000$, the results are similar to the ones presented in Table 3, where we allow only orthogonal gradient updates. The best performance in terms of ACC is obtained at $\alpha=10$ with CC=1.42x, and we report these results in Table 7.

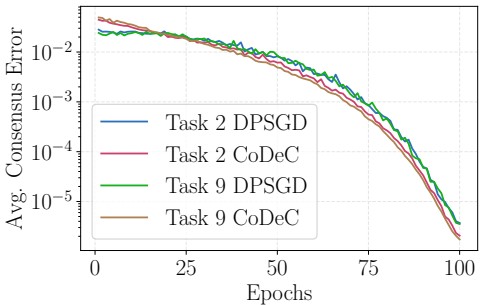

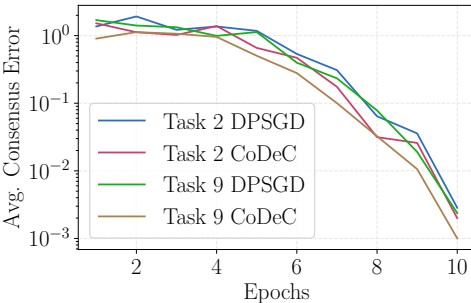

Figure 5: Average consensus error (CE) for Split CIFAR-100 (left) and Split MiniImageNet (right) over an 8 agent ring topology. Task '$\tau$' DPSGD (CoDeC) denotes CE when $\tau^{th}$ task is learned without (with) orthogonal gradient constraints.

**Consensus Error:** We also investigate the effect of taking orthogonal gradient updates upon the average consensus error, which we formally define as:

$$CE = \frac{1}{N} \sum_{i=1}^{N} \|\bar{\mathbf{x}}_k - \mathbf{x}_k^i\|^2 \qquad (13)$$

Here, $\bar{\mathbf{x}}_k$ represents the global average of the model parameters $\mathbf{x}_k^i$ at any given iteration $k$. We provide an upper bound for the consensus error in Appendix A.2. CE is a measure of the effectiveness of gossip

averaging in the decentralized learning scenario. In particular, a lower CE implies that the agents are closer to achieving a global consensus. In figure 5, we show CE with and without orthogonal updates for task 2 and 9 after each training epoch for Split CIFAR-100 and Split miniImageNet. As the training progresses, CE consistently reduces as expected. We observe that the rate of achieving consensus is similar for the two cases. In other words, CoDeC enables decentralized continual learning without hindering the gossip averaging mechanism.

## 7 Conclusion

This work proposes CoDeC, a novel communication-efficient decentralized continual learning algorithm. CoDeC enables serverless training with spatially and temporally distributed private data and mitigates catastrophic forgetting by taking gradient steps orthogonal to the gradient directions important for previous tasks. These orthogonal gradient updates, and hence the model updates, lie in a lower dimensional gradient subspace. We exploit this fact to achieve lossless communication compression without requiring any additional hyperparameters. Further, we provide theoretical insights into the convergence rate of our algorithm. Our results demonstrate that CoDeC is very effective in learning distributed continual tasks with minimal backward transfer and up to 4.8x reduced communication overhead during training.

## 8 Acknowledgments

This work was supported in part by, the Center for the Co-Design of Cognitive Systems (COCOSYS), a DARPA-sponsored JUMP center, the Semiconductor Research Corporation (SRC), the National Science Foundation, and DARPA ShELL.

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

# A    Appendix

Proofs for the lemma, theorem and corollary presented in the main paper are detailed in A.1, A.2, A.3, A.4 and A.5 sections. Details related to the network architectures and datasets used in our experiments are presented in A.7 and A.8 respectively. We list all our training hyperparameters in A.9. We also provide details about implementation of our baselines D-EWC and D-SI in A.10. Some additional results are available in A.11. We perform our experiments on a single machine with 4 NVIDIA GeForce GTX 1080 Ti GPUs. All the agents in our experiments are distributed evenly over these 4 GPUs. For instance, in the case of a 16-agent ring/torus topology, each GPU is utilized by 4 agents.

## A.1    Proof of Lemma 1

The orthogonal projection $\tilde{\mathbf{g}}^i$ of the original gradient update $\mathbf{g}^i$ with respect to GPM is obtained as:

$$\tilde{\mathbf{g}}^i = \mathbf{g}^i - (\mathbf{M}^l(\mathbf{M}^l)^T)\mathbf{g}^i \tag{14}$$

From the above equation we can write:

$$\mathbf{g}^i = \tilde{\mathbf{g}}^i + (\mathbf{M}^l(\mathbf{M}^l)^T)\mathbf{g}^i \tag{15}$$

We have:

$$\langle \mathbf{g}^i, \tilde{\mathbf{g}}^i \rangle = \langle \tilde{\mathbf{g}}^i + (\mathbf{M}^l(\mathbf{M}^l)^T)\mathbf{g}^i, \tilde{\mathbf{g}}^i \rangle = \langle \tilde{\mathbf{g}}^i, \tilde{\mathbf{g}}^i \rangle + \langle (\mathbf{M}^l(\mathbf{M}^l)^T)\mathbf{g}^i, \tilde{\mathbf{g}}^i \rangle \tag{16}$$

Since $\tilde{\mathbf{g}}^i$ and $\mathbf{M}^l(\mathbf{M}^l)^T)\mathbf{g}^i$ are orthogonal to each other:

$$\langle (\mathbf{M}^l(\mathbf{M}^l)^T)\mathbf{g}^i, \tilde{\mathbf{g}}^i \rangle = 0 \tag{17}$$

Substituting equation 17 into equation 16:

$$\langle \mathbf{g}^i, \tilde{\mathbf{g}}^i \rangle = \langle \tilde{\mathbf{g}}^i, \tilde{\mathbf{g}}^i \rangle = \|\tilde{\mathbf{g}}^i\|^2 > 0 \tag{18}$$

We make sure $\|\tilde{\mathbf{g}}^i\| \neq 0$ by using $\epsilon_{th} < 1$. From the above equation, we see that the dot product is greater than 0. This implies that if $-\mathbf{g}^i$ is in the descent direction, $-\tilde{\mathbf{g}}^i$ is also in the descent direction.

## A.2    Bounds on Consensus Error

This section provides an upper bound on the consensus error. We follow the same approach as Esfandiari et al. (2021). The update rule for our algorithm is as follows:

$$\bar{\mathbf{x}}_k = \bar{\mathbf{x}}_{k-1} - \eta \frac{1}{N} \sum_{i=1}^{N} \tilde{\mathbf{g}}_{k-1}^i \tag{19}$$

$\bar{\mathbf{x}}_k$ denotes the averaged model across all the agents at a given iteration $k$. For the rest of the analysis, the initial value will be directly set to 0. From equation 19 we have:

$$\bar{\mathbf{x}}_{k+1} - \bar{\mathbf{x}}_k = -\eta \frac{1}{N} \sum_{i=1}^{N} \tilde{\mathbf{g}}_k^i \tag{20}$$

We introduce some key notations and properties:

$$\begin{aligned}
\mathbf{Q} &= \frac{1}{N} \mathbf{1}\mathbf{1}^\top \\
\tilde{\mathbf{G}}_k &\triangleq [\tilde{\mathbf{g}}_k^1, \tilde{\mathbf{g}}_k^2, ..., \tilde{\mathbf{g}}_k^N] \\
\mathbf{X}_k &\triangleq [\mathbf{x}_k^1, \mathbf{x}_k^2, ..., \mathbf{x}_k^N] \\
\mathbf{G}_k &\triangleq [\mathbf{g}_k^1, \mathbf{g}_k^2, ..., \mathbf{g}_k^N] \\
\mathbf{H}_k &\triangleq [\nabla f_1(\mathbf{x}_k^1), \nabla f_2(\mathbf{x}_k^2), ..., \nabla f_N(\mathbf{x}_k^N)]
\end{aligned} \tag{21}$$

For all the above matrices, $\|\mathbf{A}\|_{\mathfrak{F}}^2 = \sum_{i=1}^N \|\mathbf{a}_i\|^2$, where $\mathbf{a}_i$ is the $i$-th column of the matrix $\mathbf{A}$. Thus, we obtain:

$$\|\mathbf{X}_k(\mathbf{I} - \mathbf{Q})\|_{\mathfrak{F}}^2 = \sum_{i=1}^N \|\mathbf{x}_k^i - \bar{\mathbf{x}}_k\|^2. \tag{22}$$

For each doubly stochastic matrix $\mathbf{W}$, the following properties hold true

- $\mathbf{QW} = \mathbf{WQ}$;

- $(\mathbf{I} - \mathbf{Q})\mathbf{W} = \mathbf{W}(\mathbf{I} - \mathbf{Q})$;

- For any integer $k \geq 1$, $\|(\mathbf{I} - \mathbf{Q})\mathbf{W}\|_{\mathfrak{S}} \leq (\sqrt{\rho})^k$, where $\|\cdot\|_{\mathfrak{S}}$ is the spectrum norm of a matrix.

For $N$ arbitrary real square matrices $\mathbf{A}_i, i \in \{1, 2, ..., N\}$,

$$\|\sum_{i=1}^N \mathbf{A}_i\|_{\mathfrak{F}}^2 \leq \sum_{i=1}^N \sum_{j=1}^N \|\mathbf{A}_i\|_{\mathfrak{F}} \|\mathbf{A}_j\|_{\mathfrak{F}}. \tag{23}$$

We are now ready to provide a bound on the consensus error. Since $\mathbf{X}_k = \mathbf{X}_{k-1}\mathbf{W} - \eta\tilde{\mathbf{G}}_k$ we have:

$$\mathbf{X}_k(\mathbf{I} - \mathbf{Q}) = \mathbf{X}_{k-1}(\mathbf{I} - \mathbf{Q})\mathbf{W} - \eta\tilde{\mathbf{G}}_k(\mathbf{I} - \mathbf{Q}) \tag{24}$$

Applying the above equation $k$ times we have:

$$\mathbf{X}_k(\mathbf{I} - \mathbf{Q}) = \mathbf{X}_0(\mathbf{I} - \mathbf{Q})\mathbf{W}^k - \sum_{\tau=1}^k \eta\tilde{\mathbf{G}}_\tau(\mathbf{I} - \mathbf{Q})\mathbf{W}^{k-\tau} = -\eta\sum_{\tau=1}^k \tilde{\mathbf{G}}_\tau(\mathbf{I} - \mathbf{Q})\mathbf{W}^{k-\tau} \tag{25}$$

$$\mathbb{E}\left[\left\|\mathbf{X}_k(\mathbf{I} - \mathbf{Q})\right\|_{\mathfrak{F}}^2\right] = \eta^2 \underbrace{\mathbb{E}\left[\left\|\sum_{\tau=0}^{k-1} \tilde{\mathbf{G}}_\tau(\mathbf{I} - \mathbf{Q})\mathbf{W}^{k-1-\tau}\right\|_{\mathfrak{F}}^2\right]}_{I} \tag{26}$$

We find the upper bound for term $I$.

$$
\begin{aligned}
\mathbb{E}\left[\left\|\sum_{\tau=0}^{k-1} \tilde{\mathbf{G}}_\tau(\mathbf{I} - \mathbf{Q})\mathbf{W}^{k-1-\tau}\right\|_{\mathfrak{F}}^2\right] &\overset{a}{\leq} \sum_{\tau=0}^{k-1}\sum_{\tau'=0}^{k-1} \mathbb{E}\left[\left\|\tilde{\mathbf{G}}_\tau(\mathbf{I} - \mathbf{Q})\mathbf{W}^{k-1-\tau}\right\|_{\mathfrak{F}}\left\|\tilde{\mathbf{G}}_{\tau'}(\mathbf{I} - \mathbf{Q})\mathbf{W}^{k-1-\tau'}\right\|_{\mathfrak{F}}\right] \\
&\leq \sum_{\tau=0}^{k-1}\sum_{\tau'=0}^{k-1} \rho^{(k-1-\frac{\tau+\tau'}{2})}\mathbb{E}[\|\tilde{\mathbf{G}}_\tau\|_{\mathfrak{F}}\|\tilde{\mathbf{G}}_{\tau'}\|_{\mathfrak{F}}] \overset{b}{\leq} \sum_{\tau=0}^{k-1}\sum_{\tau'=0}^{k-1} \mu^2\rho^{(k-1-\frac{\tau+\tau'}{2})}\mathbb{E}[\|\mathbf{G}_\tau\|_{\mathfrak{F}}\|\mathbf{G}_{\tau'}\|_{\mathfrak{F}}] \\
&\overset{c}{\leq} \sum_{\tau=0}^{k-1}\sum_{\tau'=0}^{k-1} \mu^2\rho^{(k-1-\frac{\tau+\tau'}{2})}\left(\frac{1}{2}\mathbb{E}[\|\mathbf{G}_\tau\|_{\mathfrak{F}}^2] + \frac{1}{2}\mathbb{E}[\|\mathbf{G}_{\tau'}\|_{\mathfrak{F}}^2]\right) \\
&= \sum_{\tau=0}^{k-1}\sum_{\tau'=0}^{k-1} \mu^2\rho^{(k-1-\frac{\tau+\tau'}{2})}\mathbb{E}[\|\mathbf{G}_\tau\|_{\mathfrak{F}}^2] \overset{d}{\leq} \frac{\mu^2}{(1-\sqrt{\rho})}\sum_{\tau=0}^{k-1}\rho^{(\frac{k-1-\tau}{2})}\mathbb{E}[\|\mathbf{G}_\tau\|_{\mathfrak{F}}^2]
\end{aligned} \tag{27}
$$

(a) follows from equation 23.
(b) follows from assumption 4.

(c) follows from the inequality $xy \leq \frac{1}{2}(x^2 + y^2)$ for any two real numbers $x, y$.

(d) is derived from $\sum_{\tau_1=0}^{k-1} \rho^{k-1-\frac{\tau_1+\tau}{2}} \leq \frac{\rho^{\frac{k-1-\tau}{2}}}{1-\sqrt{\rho}}$.

We proceed with finding the bounds for $\mathbb{E}[\|\mathbf{G}_\tau\|_{\mathfrak{F}}^2]$:

$$\mathbb{E}[\|\mathbf{G}_\tau\|_{\mathfrak{F}}^2] = \mathbb{E}[\|\mathbf{G}_\tau - \mathbf{H}_\tau + \mathbf{H}_\tau - \mathbf{H}_\tau \mathbf{Q} + \mathbf{H}_\tau \mathbf{Q}\|_{\mathfrak{F}}^2]$$

$$\leq 3\mathbb{E}[\|\mathbf{G}_\tau - \mathbf{H}_\tau\|_{\mathfrak{F}}^2] + 3\mathbb{E}[\|\mathbf{H}_\tau (I - \mathbf{Q})\|^2 \mathfrak{F}] + 3\mathbb{E}[\|\mathbf{H}_\tau \mathbf{Q}\|_{\mathfrak{F}}^2] \overset{a}{\leq} 3N\sigma^2 + 3N\delta^2 + 3\mathbb{E}[\|\frac{1}{N}\sum_{i=1}^N \nabla f_i(\mathbf{x}_\tau^i)\|^2] \quad (28)$$

(a) holds because $\mathbb{E}[\|\mathbf{H}_\tau \mathbf{Q}\|_{\mathfrak{F}}^2] \leq \mathbb{E}[\|\frac{1}{N}\sum_{i=1}^N \nabla f_i(\mathbf{x}_\tau^i)\|^2]$

Substituting (28) in (27):

$$\mathbb{E}\left[\left\|\sum_{\tau=0}^{k-1} \tilde{\mathbf{G}}_\tau (\mathbf{I} - \mathbf{Q}) \mathbf{W}^{k-1-\tau}\right\|_{\mathfrak{F}}^2\right] \leq \frac{\mu^2}{(1-\sqrt{\rho})} \sum_{\tau=0}^{k-1} \rho^{(\frac{k-1-\tau}{2})} \left[3N\sigma^2 + 3N\delta^2 + 3\mathbb{E}[\|\frac{1}{N}\sum_{i=1}^N \nabla f_i(\mathbf{x}_\tau^i)\|^2]\right]$$

$$\leq \frac{3N\mu^2(\sigma^2 + \delta^2)}{(1-\sqrt{\rho})^2} + \frac{3N\mu^2}{(1-\sqrt{\rho})} \sum_{\tau=0}^{k-1} \rho^{(\frac{k-1-\tau}{2})} \mathbb{E}[\|\frac{1}{N}\sum_{i=1}^N \nabla f_i(\mathbf{x}_\tau^i)\|^2] \quad (29)$$

Substituting (29) into the main inequality (26):

$$\mathbb{E}\left[\left\|\mathbf{X}_k(\mathbf{I} - \mathbf{Q})\right\|_{\mathfrak{F}}^2\right] \leq \eta^2\mu^2\left(\frac{3N\sigma^2}{(1-\sqrt{\rho})^2} + \frac{3N\delta^2}{(1-\sqrt{\rho})^2}\right) + \frac{3N\eta^2\mu^2}{(1-\sqrt{\rho})} \sum_{\tau=0}^{k-1} \rho^{(\frac{k-1-\tau}{2})} \mathbb{E}[\|\frac{1}{N}\sum_{i=1}^N \nabla f_i(\mathbf{x}_\tau^i)\|^2] \quad (30)$$

Summing over $k \in \{1, \ldots, K-1\}$ and noting that $\mathbb{E}\left[\left\|\mathbf{X}_0(\mathbf{I} - \mathbf{Q})\right\|_{\mathfrak{F}}^2\right] = 0$:

$$\sum_{k=1}^{K-1} \mathbb{E}\left[\left\|\mathbf{X}_k(\mathbf{I} - \mathbf{Q})\right\|_{\mathfrak{F}}^2\right] \leq CK + \frac{3N\eta^2\mu^2}{(1-\sqrt{\rho})} \sum_{k=1}^{K-1}\sum_{\tau=0}^{k-1} \rho^{(\frac{k-1-\tau}{2})} \mathbb{E}[\|\frac{1}{N}\sum_{i=1}^N \nabla f_i(\mathbf{x}_\tau^i)\|^2] \leq$$

$$CK + \frac{3N\eta^2\mu^2}{(1-\sqrt{\rho})} \sum_{k=0}^{K-1} \frac{1 - \rho^{(\frac{K-1-k}{2})}}{1-\sqrt{\rho}} \mathbb{E}[\|\frac{1}{N}\sum_{i=1}^N \nabla f_i(\mathbf{x}_k^i)\|^2] \leq CK + \frac{3N\eta^2\mu^2}{(1-\sqrt{\rho})} \sum_{k=0}^{K-1} \mathbb{E}[\|\frac{1}{N}\sum_{i=1}^N \nabla f_i(\mathbf{x}_k^i)\|^2] \quad (31)$$

$$where \ C = \eta^2\mu^2\left(\frac{3N\sigma^2 + 3N\delta^2}{(1-\sqrt{\rho})^2}\right)$$

Dividing both sides by $N$:

$$\sum_{k=1}^{K-1} \frac{1}{N}\mathbb{E}\left[\left\|\mathbf{X}_k(\mathbf{I} - \mathbf{Q})\right\|_{\mathfrak{F}}^2\right] \leq \eta^2\mu^2\left(\frac{3\sigma^2}{(1-\sqrt{\rho})^2} + \frac{3\delta^2}{(1-\sqrt{\rho})^2}\right)K + \frac{3\eta^2\mu^2}{(1-\sqrt{\rho})} \sum_{k=0}^{K-1} \mathbb{E}[\|\frac{1}{N}\sum_{i=1}^N \nabla f_i(\mathbf{x}_k^i)\|^2] \quad (32)$$

This directly implies:

$$\sum_{k=0}^{K-1} \frac{1}{N}\sum_{i=1}^N \mathbb{E}\left[\left\|\bar{\mathbf{x}}_k - \mathbf{x}_k^i\right\|^2\right] \leq \eta^2\mu^2\left(\frac{3\sigma^2}{(1-\sqrt{\rho})^2} + \frac{3\delta^2}{(1-\sqrt{\rho})^2}\right)K + \frac{3\eta^2\mu^2}{(1-\sqrt{\rho})} \sum_{k=0}^{K-1} \mathbb{E}[\|\frac{1}{N}\sum_{i=1}^N \nabla f_i(\mathbf{x}_k^i)\|^2] \quad (33)$$

### A.3 Proof for Theorem 2

When $\mathcal{F}$ is $L$-smooth, we have:

$$\mathbb{E}[\mathcal{F}(\bar{\mathbf{x}}_{k+1})] \leq \mathbb{E}[\mathcal{F}(\bar{\mathbf{x}}_k)] + \underbrace{\mathbb{E}[\langle \nabla \mathcal{F}(\bar{\mathbf{x}}_k), \bar{\mathbf{x}}_{k+1} - \bar{\mathbf{x}}_k \rangle]}_{I} + \frac{L}{2}\mathbb{E}[\|\bar{\mathbf{x}}_{k+1} - \bar{\mathbf{x}}_k\|^2] \tag{34}$$

We proceed by analysing $I$:

$$\mathbb{E}[\langle \nabla \mathcal{F}(\bar{\mathbf{x}}_k), \bar{\mathbf{x}}_{k+1} - \bar{\mathbf{x}}_k \rangle] = \mathbb{E}[\langle \nabla \mathcal{F}(\bar{\mathbf{x}}_k), -\eta\Big(\frac{1}{N}\sum_{i=1}^{N}\tilde{\mathbf{g}}_k^i\Big)\rangle] \tag{35}$$

$$\mathbb{E}[\langle \nabla \mathcal{F}(\bar{\mathbf{x}}_k), -\eta\Big(\frac{1}{N}\sum_{i=1}^{N}\tilde{\mathbf{g}}_k^i\Big)\rangle] = \mathbb{E}[\langle \nabla \mathcal{F}(\bar{\mathbf{x}}_k), -\eta\Big(\frac{1}{N}\sum_{i=1}^{N}\tilde{\mathbf{g}}_k^i - \mathbf{g}_k^i + \mathbf{g}_k^i\Big)\rangle]$$

$$= \underbrace{\mathbb{E}[\langle \nabla \mathcal{F}(\bar{\mathbf{x}}_k), -\eta\Big(\frac{1}{N}\sum_{i=1}^{N}\tilde{\mathbf{g}}_k^i - \mathbf{g}_k^i\Big)\rangle]}_{II} + \underbrace{\mathbb{E}[\langle \nabla \mathcal{F}(\bar{\mathbf{x}}_k), -\eta\Big(\frac{1}{N}\sum_{i=1}^{N}\mathbf{g}_k^i\Big)\rangle]}_{III} \tag{36}$$

We first analyse $II$:

$$-\eta\mathbb{E}[\langle \nabla \mathcal{F}(\bar{\mathbf{x}}_k), \frac{1}{N}\sum_{i=1}^{N}\big(\tilde{\mathbf{g}}_k^i - \mathbf{g}_k^i\big)\rangle] \leq \frac{1}{2L}\mathbb{E}[\|\nabla \mathcal{F}(\bar{\mathbf{x}}_k)\|^2] + \frac{L\eta^2}{2}\mathbb{E}[\|\frac{1}{N}\sum_{i=1}^{N}(\tilde{\mathbf{g}}_k^i - \mathbf{g}_k^i)\|^2] \tag{37}$$

This holds as $\langle \mathbf{a}, \mathbf{b}\rangle \leq \frac{1}{2}\|\mathbf{a}\|^2 + \frac{1}{2}\|\mathbf{b}\|^2$.
Analysing $III$:

$$\mathbb{E}\left[\langle \nabla \mathcal{F}(\bar{\mathbf{x}}_k), -\eta\Big(\frac{1}{N}\sum_{i=1}^{N}\mathbf{g}_k^i\Big)\rangle\right] = -\eta\mathbb{E}\left[\langle \nabla \mathcal{F}(\bar{\mathbf{x}}_k), \frac{1}{N}\sum_{i=1}^{N}\nabla f_i(\mathbf{x}_k^i)\rangle\right] \tag{38}$$

With the aid of the equity $\langle \mathbf{a}, \mathbf{b}\rangle = \frac{1}{2}[\|\mathbf{a}\|^2 + \|\mathbf{b}\|^2 - \|\mathbf{a} - \mathbf{b}\|^2]$, we have :

$$\langle \nabla \mathcal{F}(\bar{\mathbf{x}}_k), \frac{1}{N}\sum_{i=1}^{N}\nabla f_i(\mathbf{x}_k^i)\rangle = \frac{1}{2}\left(\|\nabla \mathcal{F}(\bar{\mathbf{x}}_k)\|^2 + \|\frac{1}{N}\sum_{i=1}^{N}\nabla f_i(\mathbf{x}_k^i)\|^2 - \underbrace{\|\nabla \mathcal{F}(\bar{\mathbf{x}}_k) - \frac{1}{N}\sum_{i=1}^{N}\nabla f_i(\mathbf{x}_k^i)\|^2}_{\star}\right) \tag{39}$$

Analysing $\star$:

$$\|\nabla \mathcal{F}(\bar{\mathbf{x}}_k) - \frac{1}{N}\sum_{i=1}^{N}\nabla f_i(\mathbf{x}_k^i)\|^2 = \|\frac{1}{N}\sum_{i=1}^{N}\nabla f_i(\bar{\mathbf{x}}_k) - \frac{1}{N}\sum_{i=1}^{N}\nabla f_i(\mathbf{x}_k^i)\|^2$$

$$\leq \frac{1}{N}\sum_{i=1}^{N}\|\nabla f_i(\bar{\mathbf{x}}_k) - \nabla f_i(\mathbf{x}_k^i)\|^2 \leq \frac{1}{N}\sum_{i=1}^{N}L^2\|\bar{\mathbf{x}}_k - \mathbf{x}_k^i\|^2 \tag{40}$$

Substituting (40) back into (39), we have:

$$\langle \nabla \mathcal{F}(\bar{\mathbf{x}}_k), \frac{1}{N}\sum_{i=1}^{N}\nabla f_i(\mathbf{x}_k^i)\rangle \geq \frac{1}{2}\left(\|\nabla \mathcal{F}(\bar{\mathbf{x}}_k)\|^2 + \|\frac{1}{N}\sum_{i=1}^{N}\nabla f_i(\mathbf{x}_k^i)\|^2 - L^2\frac{1}{N}\sum_{i=1}^{N}\|\bar{\mathbf{x}}_k - \mathbf{x}_k^i\|^2\right) \tag{41}$$

Substituting (37) and (41) into (36), and (36) into (35):

$$\mathbb{E}[\langle \nabla \mathcal{F}(\bar{\mathbf{x}}_k), \bar{\mathbf{x}}_{k+1} - \bar{\mathbf{x}}_k \rangle] \leq \left( \frac{1}{2L} - \frac{\eta}{2} \right) \mathbb{E}[\|\nabla \mathcal{F}(\bar{\mathbf{x}}_k)\|^2] + \frac{L\eta^2}{2} \mathbb{E}[\| \frac{1}{N} \sum_{i=1}^{N} (\tilde{\mathbf{g}}_k^i - \mathbf{g}_k^i)\|^2]$$
$$- \frac{\eta}{2} \left( \mathbb{E}[\| \frac{1}{N} \sum_{i=1}^{N} \nabla f_i(\mathbf{x}_k^i)\|^2] - L^2 \mathbb{E}[\frac{1}{N} \sum_{i=1}^{N} \|\bar{\mathbf{x}}_k - \mathbf{x}_k^i\|^2] \right) \tag{42}$$

From equation (20), we have:

$$\mathbb{E}[\|\bar{\mathbf{x}}_{k+1} - \bar{\mathbf{x}}_k\|^2] = \eta^2 \mathbb{E}[\| \frac{1}{N} \sum_{i=1}^{N} \tilde{\mathbf{g}}_k^i \|^2]. \tag{43}$$

Substituting (42) and (43) in (34):

$$\mathbb{E}[\mathcal{F}(\bar{\mathbf{x}}_{k+1})] \leq \mathbb{E}[\mathcal{F}(\bar{\mathbf{x}}_k)] + \left( \frac{1}{2L} - \frac{\eta}{2} \right) \mathbb{E}[\|\nabla \mathcal{F}(\bar{\mathbf{x}}_k)\|^2] + \frac{L\eta^2}{2} \mathbb{E}[\| \frac{1}{N} \sum_{i=1}^{N} (\tilde{\mathbf{g}}_k^i - \mathbf{g}_k^i)\|^2]$$
$$- \frac{\eta}{2} \mathbb{E}[\| \frac{1}{N} \sum_{i=1}^{N} \nabla f_i(\mathbf{x}_k^i)\|^2] + \frac{\eta L^2}{2} \mathbb{E}[\frac{1}{N} \sum_{i=1}^{N} \|\bar{\mathbf{x}}_k - \mathbf{x}_k^i\|^2] + \frac{\eta^2 L}{2} \mathbb{E}[\| \frac{1}{N} \sum_{i=1}^{N} \tilde{\mathbf{g}}_k^i \|^2] \tag{44}$$

Rearranging the terms and dividing by $C_1 = \left( \frac{\eta}{2} - \frac{1}{2L} \right) > 0$ to find the bound for $\mathbb{E}[\|\nabla \mathcal{F}(\bar{\mathbf{x}}_k)\|^2]$:

$$\mathbb{E}[\|\nabla \mathcal{F}(\bar{\mathbf{x}}_k)\|^2] \leq \frac{1}{C_1} \left( \mathbb{E}[\mathcal{F}(\bar{\mathbf{x}}_k)] - \mathbb{E}[\mathcal{F}(\bar{\mathbf{x}}_{k+1})] \right) + C_2 \left( \underbrace{\mathbb{E}[\| \frac{1}{N} \sum_{i=1}^{N} (\tilde{\mathbf{g}}_k^i - \mathbf{g}_k^i)\|^2] + \mathbb{E}[\| \frac{1}{N} \sum_{i=1}^{N} \tilde{\mathbf{g}}_k^i \|^2]}_{\star} \right)$$
$$+ C_3 \, \mathbb{E}[\frac{1}{N} \sum_{i=1}^{N} \|\bar{\mathbf{x}}_k - \mathbf{x}_k^i\|^2] - C_4 \, \mathbb{E}[\| \frac{1}{N} \sum_{i=1}^{N} \nabla f_i(\mathbf{x}_k^i)\|^2] \tag{45}$$

*where* $C_2 = L\eta^2/2C_1, C_3 = L^2\eta/2C_1, C_4 = \eta/2C_1$.

We first analyze $\star$:

$$\mathbb{E}[\| \frac{1}{N} \sum_{i=1}^{N} (\tilde{\mathbf{g}}_k^i - \mathbf{g}_k^i)\|^2] + \mathbb{E}[\| \frac{1}{N} \sum_{i=1}^{N} \tilde{\mathbf{g}}_k^i \|^2] = \frac{1}{N^2} \mathbb{E}[\| \sum_{i=1}^{N} (\tilde{\mathbf{g}}_k^i - \mathbf{g}_k^i)\|^2 + \| \sum_{i=1}^{N} \tilde{\mathbf{g}}_k^i \|^2]$$
$$\overset{a}{=} \frac{1}{N^2} \mathbb{E}[\| \sum_{i=1}^{N} (\mathbf{M}\mathbf{M^T}\mathbf{g}_k^i)\|^2 + \| \sum_{i=1}^{N} ((\mathbf{I} - \mathbf{M}\mathbf{M^T})\mathbf{g}_k^i)\|^2] \overset{b}{=} \frac{1}{N^2} \mathbb{E}[\|\mathbf{M}\mathbf{M^T} \sum_{i=1}^{N} (\mathbf{g}_k^i)\|^2 + \|(\mathbf{I} - \mathbf{M}\mathbf{M^T}) \sum_{i=1}^{N} (\mathbf{g}_k^i)\|^2]$$
$$= \mathbb{E}[\| \sum_{i=1}^{N} \frac{1}{N} \mathbf{g}_k^i \|^2] \overset{c}{\leq} \left( \frac{\sigma^2}{N} + \mathbb{E}\left[ \left\| \frac{1}{N} \sum_{i=1}^{N} \nabla f_i(\mathbf{x}^i) \right\|^2 \right] \right) \tag{46}$$

(a) follows from the fact that $\tilde{\mathbf{g}}_k^i$ is an orthogonal projection of $\mathbf{g}_k^i$, and it is defined by the GPM matrix $\mathbf{M}$.

(b) follows from all agents having the same GPM matrix $\mathbf{M}$

(c) is the conclusion of Lemma 1 in Yu et al. (2019).

Substituting (46) into (45) and summing over $k \in \{0, 1, \ldots, K-1\}$:

$$
\begin{aligned}
\sum_{k=0}^{K-1} \mathbb{E}\left[\|\nabla \mathcal{F}\left(\bar{\mathbf{x}}_k\right)\|^2\right] &\leq \frac{1}{C_1}\left(\mathbb{E}\left[\mathcal{F}\left(\bar{\mathbf{x}}_0\right) - \mathcal{F}\left(\bar{\mathbf{x}}_k\right)\right]\right) + C_2 \sum_{k=0}^{K-1}\left(\frac{\sigma^2}{N} + \mathbb{E}\left[\left\|\frac{1}{N}\sum_{i=1}^{N}\nabla f_i(\mathbf{x}_k^i)\right\|^2\right]\right) \\
&+ C_3 \sum_{k=0}^{K-1}\mathbb{E}\left[\frac{1}{N}\sum_{i=1}^{N}\left\|\bar{\mathbf{x}}_k - \mathbf{x}_k^i\right\|^2\right] - C_4 \sum_{k=0}^{K-1}\mathbb{E}\left[\left\|\frac{1}{N}\sum_{i=1}^{N}\nabla f_i(\mathbf{x}_k^i)\right\|^2\right]
\end{aligned}
\tag{47}
$$

Dividing both sides by $K$:

$$
\begin{aligned}
\frac{1}{K}\sum_{k=0}^{K-1} \mathbb{E}\left[\|\nabla \mathcal{F}\left(\bar{\mathbf{x}}_k\right)\|^2\right] &\leq \frac{1}{C_1 K}\left(\mathbb{E}\left[\mathcal{F}\left(\bar{\mathbf{x}}_0\right) - \mathcal{F}^*\right]\right) + C_2 \frac{\sigma^2}{N} + C_2 \sum_{k=0}^{K-1}\frac{1}{K}\left(\mathbb{E}\left[\left\|\frac{1}{N}\sum_{i=1}^{N}\nabla f_i(\mathbf{x}_k^i)\right\|^2\right]\right) \\
&+ \frac{C_3}{K} \sum_{k=0}^{K-1}\mathbb{E}\left[\frac{1}{N}\sum_{i=1}^{N}\left\|\bar{\mathbf{x}}_k - \mathbf{x}_k^i\right\|^2\right] - C_4 \sum_{k=0}^{K-1}\frac{1}{K}\mathbb{E}\left[\left\|\frac{1}{N}\sum_{i=1}^{N}\nabla f_i(\mathbf{x}_k^i)\right\|^2\right]
\end{aligned}
\tag{48}
$$

Using equation 33 in the above equation, we have:

$$
\begin{aligned}
\frac{1}{K}\sum_{k=0}^{K-1} \mathbb{E}\left[\|\nabla \mathcal{F}\left(\bar{\mathbf{x}}_k\right)\|^2\right] &\leq \frac{1}{C_1 K}\left(\mathbb{E}\left[\mathcal{F}\left(\bar{\mathbf{x}}_0\right) - \mathcal{F}^*\right]\right) + C_2 \frac{\sigma^2}{N} + C_2 \sum_{k=0}^{K-1}\frac{1}{K}\left(\mathbb{E}\left[\left\|\frac{1}{N}\sum_{i=1}^{N}\nabla f_i(\mathbf{x}_k^i)\right\|^2\right]\right) \\
&+ \frac{C_3}{K}\left[\eta^2\mu^2\left(\frac{3\sigma^2}{(1-\sqrt{\rho})^2} + \frac{3\delta^2}{(1-\sqrt{\rho})^2}\right)K + \frac{3\eta^2\mu^2}{(1-\sqrt{\rho})}\sum_{k=0}^{K-1}\mathbb{E}[\|\frac{1}{N}\sum_{i=1}^{N}\nabla f_i(\mathbf{x}_k^i)\|^2]\right] \\
&- C_4 \sum_{k=0}^{K-1}\frac{1}{K}\mathbb{E}\left[\left\|\frac{1}{N}\sum_{i=1}^{N}\nabla f_i(\mathbf{x}_k^i)\right\|^2\right]
\end{aligned}
\tag{49}
$$

Rearranging the terms:

$$
\begin{aligned}
\frac{1}{K}\sum_{k=0}^{K-1} \mathbb{E}\left[\|\nabla \mathcal{F}\left(\bar{\mathbf{x}}_k\right)\|^2\right] &\leq \frac{1}{C_1 K}\left(\mathbb{E}\left[\mathcal{F}\left(\bar{\mathbf{x}}_0\right) - \mathcal{F}^*\right]\right) + C_2 \frac{\sigma^2}{N} + C_3\,\eta^2\mu^2\left(\frac{3\sigma^2}{(1-\sqrt{\rho})^2} + \frac{3\delta^2}{(1-\sqrt{\rho})^2}\right) \\
&+ \left(C_2 + \frac{3C_3\eta^2\mu^2}{(1-\sqrt{\rho})} - C_4\right)\left(\frac{1}{K}\sum_{k=0}^{K-1}\mathbb{E}\left[\left\|\frac{1}{N}\sum_{i=1}^{N}\nabla f_i(\mathbf{x}_k^i)\right\|^2\right]\right)
\end{aligned}
\tag{50}
$$

When $\left(C_2 + \frac{3C_3\eta^2\mu^2}{(1-\sqrt{\rho})} - C_4\right) \leq 0$, we have:

$$
\frac{1}{K}\sum_{k=0}^{K-1} \mathbb{E}\left[\|\nabla \mathcal{F}\left(\bar{\mathbf{x}}_k\right)\|^2\right] \leq \frac{1}{C_1 K}\left(\mathbb{E}\left[\mathcal{F}\left(\bar{\mathbf{x}}_0\right) - \mathcal{F}^*\right]\right) + C_2 \frac{\sigma^2}{N} + C_3\,\eta^2\mu^2\left(\frac{3\sigma^2}{(1-\sqrt{\rho})^2} + \frac{3\delta^2}{(1-\sqrt{\rho})^2}\right)
\tag{51}
$$

## A.4 Discussion on the Step Size

Recall the condition $C1 > 0$. This implies $\eta > \frac{1}{L}$.
The condition for equation (51) to be true is $\left(C_2 + \frac{3C_3\eta^2\mu^2}{(1-\sqrt{\rho})} - C_4\right) \leq 0$. Therefore, we have:

$$
\frac{3L^2\eta^2\mu^2}{(1-\sqrt{\rho})} + \eta L - 1 \leq 0
\tag{52}
$$

Solving this inequality, combining the fact that $\eta > 0$, we have then the specific form of $\eta^*$:

$$
\eta^* = \frac{\sqrt{(1-\sqrt{\rho})^2 + 12\mu^2} - (1-\sqrt{\rho})}{6L\mu^2}
\tag{53}
$$

Hence, the step size $\eta$ is defined as

$$\frac{1}{L} < \eta \le \frac{\sqrt{(1 - \sqrt{\rho})^2 + 12\mu^2} - (1 - \sqrt{\rho})}{6L\mu^2} \tag{54}$$

## A.5 Proof for Corollary 3

According to equation (51), on the right hand side, there are three terms with different coefficients with respect to the step size $\eta$. We separately investigate each term:

$\eta = \mathcal{O}\left(\sqrt{\frac{N}{K}}\right)$ implies $C_1 = \mathcal{O}\left(\sqrt{\frac{N}{K}}\right)$. Therefore for the first term:

$$\frac{\mathcal{F}(\bar{\mathbf{x}}_0) - \mathcal{F}^*}{C_1 K} = \mathcal{O}\left(\frac{1}{\sqrt{NK}}\right) \tag{55}$$

For the second term:

$$\frac{C_2}{N} = \mathcal{O}\left(\frac{1}{N}\sqrt{\frac{N}{K}}\right) = \mathcal{O}\left(\frac{1}{\sqrt{NK}}\right) \tag{56}$$

For the third term:

$$\eta^2 C_3 = \mathcal{O}\left(\frac{N}{K}\right) \tag{57}$$

By omitting $N$ in non-dominant terms, there exists a constant $C > 0$ such that the overall convergence rate is as follows:

$$\frac{1}{K}\sum_{k=0}^{K-1} \mathbb{E}\left[\|\nabla\mathcal{F}(\bar{\mathbf{x}}_k)\|^2\right] \le C\left(\frac{1}{\sqrt{NK}} + \frac{1}{K}\right), \tag{58}$$

which suggests when $N$ is fixed and $K$ is sufficiently large, CoDeC enables the convergence rate of $O(\frac{1}{\sqrt{NK}})$.

## A.6 Proof for $\mathbf{q}_k^i$ Lying in RGS

Here, we prove that $\mathbf{q}_k^i$ from line 9 in algorithm 1 lies in RGS.

Line 8 (Algorithm 1): $\mathbf{x}_{k+1}^i = \mathbf{x}_{(k+\frac{1}{2})}^i + \sum_{j\in\mathcal{N}(i)} w_{ij}(\hat{\mathbf{x}}_k^j - \mathbf{x}_k^i)$

Line 9 (Algorithm 1): $\mathbf{q}_k^i = \mathbf{x}_{k+1}^i - \mathbf{x}_k^i$

After simplifying $\mathbf{q}_k^i$:

$$\mathbf{q}_k^i = \mathbf{x}_{(k+\frac{1}{2})}^i + \sum_{j\in\mathcal{N}(i)} w_{ij}(\hat{\mathbf{x}}_k^j - \mathbf{x}_k^i) - \mathbf{x}_k^i \tag{59}$$

From line 7 in Algorithm 1 we have, $\mathbf{x}_{(k+\frac{1}{2})}^i = \mathbf{x}_k^i - \eta\tilde{\mathbf{g}}_k^i$

Substituting $\mathbf{x}_{(k+\frac{1}{2})}^i$ in equation 59, $\mathbf{q}_k^i = -\eta\tilde{\mathbf{g}}_k^i + \sum_{j\in\mathcal{N}(i)} w_{ij}(\hat{\mathbf{x}}_k^j - \mathbf{x}_k^i)$

Now, we know that $\tilde{\mathbf{g}}_k^i$ is the orthogonal gradient component of $\mathbf{g}_k^i$, and hence it lies in RGS subspace. If the gossip update component i.e. $\sum_{j\in\mathcal{N}(i)} w_{ij}(\hat{\mathbf{x}}_k^j - \mathbf{x}_k^i)$ also lies in RGS, we can conclude that $\mathbf{q}_k^i$ lies in RGS by linearity.

We prove this using the linearity property of the vector spaces and induction.

Say $\alpha_k^i = \sum_j w_{ij}(\hat{x}_k^j - x_k^i) = \sum_j w_{ij}(x_k^j - x_k^i)$. $\alpha_0^i$ lies in RGS as $x_0^j = x_0^i$ (synchronized initialization). Now, $\alpha_{k+1}^i = \sum_j w_{ij}(\alpha_k^j - \eta(\tilde{g}_k^j - \tilde{g}_k^i))$.

GPM update ensures that gradients $\tilde{g}_k^j$ and $\tilde{g}_k^i$ lie in RGS and from induction, we have $\alpha_k^j$ to lie in RGS. From linearity, we conclude that $\alpha_{k+1}^i$ lies in RGS for any agent $i$. Hence, $\mathbf{q}_k^i$ also lies in RGS.

### A.7   Network Architecture

- **AlexNet-like architecture**: For our experiments, we scale the output channels in each layer of the architecture used in Serrà et al. (2018). The network consists of 3 convolutional layers of 16, 32, and 64 filters with $4 \times 4$, $3 \times 3$, and $2 \times 2$ kernel sizes, respectively and 2 fully connected layers of 512 units each. A $2 \times 2$ max-pooling layer follows the convolutional layers. Rectified linear units are used as activations. Dropout of 0.2 is used for the first two layers and 0.5 for the rest of the layers.

- **Reduced ResNet18 architecture**: This is similar to the architecture used by Lopez-Paz & Ranzato (2017). We replace the $4 \times 4$ average-pooling layer with a $2 \times 2$ layer. For experiments with miniImageNet, we use convolution with stride 2 in the first layer.

All the networks use ReLU in the hidden units and softmax with cross entropy loss in the final layer.

### A.8   Datasets

Table 8 and 9 provide the details related to the datasets used in our experiments. For MedMNIST-5 dataset statistics, please refer to table 2 in Yang et al. (2021). The training samples/tasks are independently and identically distributed (IID) across agents without any data overlap. For instance, for a graph size of 4 agents, each agent has $5000/4 = 1250$ training samples for a particular task in Split CIFAR-100.

Table 8: Dataset Statistics for Split CIFAR-100 and Split-miniImageNet

|  | Split CIFAR-100 | Split miniImageNet |
|---|---|---|
| num. of tasks | 10 | 20 |
| input size | $3 \times 32 \times 32$ | $3 \times 84 \times 84$ |
| # Classes/task | 10 | 5 |
| # Training samples/tasks | 5,000 | 2,500 |
| # Test samples/tasks | 1,000 | 500 |

Table 9: 5-Datasets Statistics

|  | CIFAR-10 | MNIST | SVHN | Fashion MNIST | notMNIST |
|---|---|---|---|---|---|
| Classes | 10 | 10 | 10 | 10 | 10 |
| # Training samples/tasks | 50,000 | 60,000 | 73,257 | 60,000 | 16,853 |
| # Test samples/tasks | 10,000 | 10,000 | 26,032 | 10,000 | 1,873 |

### A.9   Hyperparameters

All our experiments were run for three randomly chosen seeds. We decay the learning rate by a factor of 10 after 50% and 75% of the training, unless mentioned otherwise. For Split CIFAR-100, we use a mini-batch size of 22 per agent, and we run all our experiments for a total of 100 epochs for each task. For Split MiniImageNet, we use a mini-batch size of 10 per agent and 10 epochs for each task. For 5-Datasets and MedMNIST-5, we use a mini-batch size of 32 per agent, and 50 epochs for each task. We list additional hyperparameters in Table 10. For D-SI, in case of Split CIFAR-100 we use a lower learning rate of 0.0005 for the last 2 tasks when number of agents is 8 or 16. Similarly for 5-Datasets, we lower the learning rate to 0.001 after the 1st task. This tuning is required for the training to converge in case of D-SI. The average consensus error plots shown in figure 5 in the main paper were obtained with a cosine annealing based learning rate scheduling instead of the step decay mentioned earlier.

Table 10: List of hyperparameters for the baselines and our approach. 'lr' represents initial learning rate. 'incre' represents how the $\epsilon_{th}$ is incremented for each new task.

| Dataset | Setup | Hyperparameters |
|---|---|---|
| | D-SI | lr: 0.001, $c$: 0.1 |
| Split CIFAR-100 | D-EWC | lr: 0.05, $\lambda$: 5000 |
| | CoDeC | lr: 0.01, $\epsilon_{th}$: 0.97, incre: 0.003 |
| | D-SI | lr: 0.001, $c$: 0.3 |
| Split MiniImageNet | D-EWC | lr: 0.03, $\lambda$: 5000 |
| | CoDeC | lr: 0.1, $\epsilon_{th}$: 0.985, incre: 0.0003 |
| | D-SI | lr: 0.01, $c$: 0.1 |
| 5-Datasets | D-EWC | lr: 0.03, $\lambda$: 5000 |
| | CoDeC | lr: 0.1, $\epsilon_{th}$: 0.965, incre: 0 |
| | D-SI | lr: 0.001, $c$: 0.5 |
| MedMNIST-5 | D-EWC | lr: 0.001, $\lambda$: 5000 |
| | CoDeC | lr: 0.001, $\epsilon_{th}$: 0.99, incre: 0 |

## A.10   Baseline Implementation

Algorithm 2 and 3 demonstrate the flow of D-EWC and D-SI respectively, the baselines which extend EWC (Kirkpatrick et al., 2017) and SI (Zenke et al., 2017) to a decentralized setting. The loss function minimized in D-EWC is of the form $\tilde{f}_{\tau,i}(d^i_{\tau,k}; \mathbf{x}^i_k)$ shown in line 5, algorithm 2. Here, $\lambda$ is a regularization coefficient which signifies the importance given to the past tasks. $\mathbf{x}^{i,l}_k$ and $\mathbf{x}^{i,l}_{\tau-1}$ represent model parameters for a particular layer $l$. Unlike CoDeC, here we generate the Fisher matrix $\mathcal{F}^i$ at each agent and then do a global averaging step before utilizing it for continually learning the next task. We do so because D-EWC performs best when entire training data is used to generate the Fisher matrix.

SI aims to minimize the loss function $\tilde{f}_{\tau,i}(d^i_{\tau,k}; \mathbf{x}^i_k)$ of the form shown in line 5, algorithm 3. Here, $c$ is a dimensionless strength parameter which signifies the importance given to the past tasks. SI computes $\omega$ in an online manner, which is used to update $\Omega$ at the end of each task. $\Omega$ is an importance estimate, which scales the per-parameter regularization strength. $\Omega$ is generated at each agent and then globally averaged before utilizing it for the next task.

STL and D-STL provide upper bounds on performance. These are not continual learning techniques and may not be feasible in resource-constrained environments as they requires excessive number of model parameters.

## A.11   Additional Results

**Continual Learning Statistics Aggregation:** Additional results in Table 11 demonstrate the impact of broadcast and all-gather for continual learning statistics aggregation.

**Task-wise CC:** We present additional results for task-wise CC, similar to figure 3 in the paper. Figure 3(a) demonstrates that task-wise CC ranges from 1.2x to 1.8x for Split miniImageNet. Figure 3(b) shows task-wise CC ranging from 1.2x to 4.45x for Split CIFAR-100.

**Training Loss vs Epochs with and without compression:** We present some results to emphasize the lossless nature of our proposed communication compression scheme. Figure 7 shows training loss after each epoch for a particular agent for task 2 and 9 in Split CIFAR-100 sequence with and without compression. The convergence rate of the training loss is not affected by applying the proposed compression scheme.

Table 11: Impact of all-gather and broadcast technique for continual learning over 8 agent directed ring.

| Dataset | Setup | ACC | BWT |
|---|---|---|---|
| Split CIFAR-100 | D-SI (broadcast) | 27.2 | -14.14 |
| | D-SI (all-gather) | 39.36 | -1.98 |
| | D-EWC (broadcast) | 46.66 | -0.06 |
| | D-EWC (all-gather) | 50.46 | 0.42 |
| | CoDeC(broadcast) | 53.13 | -1.00 |
| | CoDeC (all-gather) | 52.7 | -0.39 |
| 5-Datasets | D-SI (broadcast) | 37.57 | -43.65 |
| | D-SI (all-gather) | 80.23 | -2.70 |
| | D-EWC (broadcast) | 71.54 | -0.12 |
| | D-EWC (all-gather) | 85.9 | -0.92 |
| | CoDeC(broadcast) | 86.56 | -4.38 |
| | CoDeC (all-gather) | 86.46 | -4.00 |

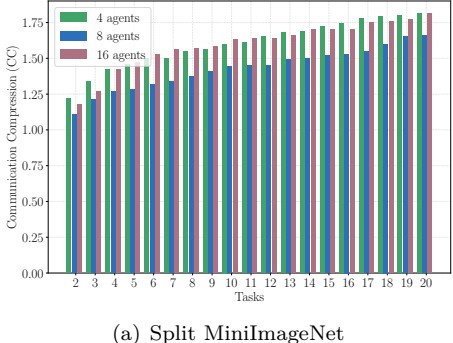

(a) Split MiniImageNet

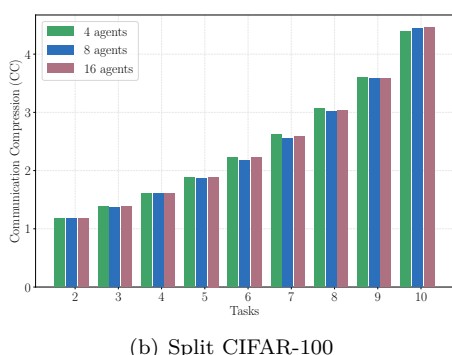

(b) Split CIFAR-100

Figure 6: Task-wise CC for (a) Split MiniImageNet and (b) Split CIFAR-100 over a directed ring topology.

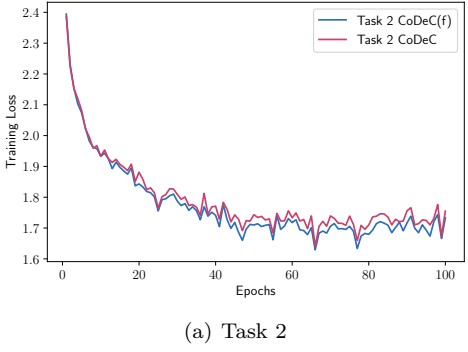

(a) Task 2

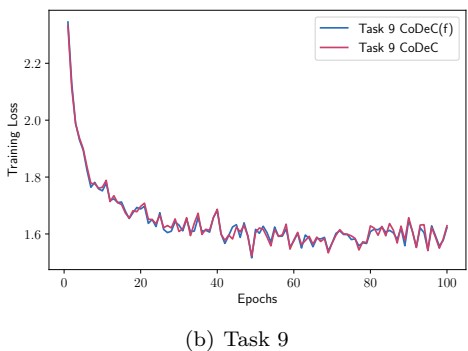

(b) Task 9

Figure 7: Training loss vs epochs for (a) task 2 and (b) task 9 in Split CIFAR-100 sequence with CoDeC(f) and CoDeC using AlexNet over a directed ring with 8 agents

**CoDeC vs SKILL:** Each agent in SKILL (Ge et al., 2023) uses a common pre-trained frozen backbone built-in at initialization so that only the last layer (or head) and a unique set of bias parameters are learned for each task. After training, these bias parameters and task-specific heads are shared among agents via a fully connected graph. However, each SKILL agent is independently learning a different task at a given time, lacking the concept of collaborative learning as demonstrated in CoDeC. The data for each task is spatially

distributed in our setup, while SKILL assumes access to the entire data available in a centralized manner at a single agent. Hence, in CoDeC it becomes essential to communicate with peers during training. Moreover, we do not assume access to a pre-trained model at initialization, and update the weights for each task while mitigating forgetting. Unlike SKILL where the number of learned parameters increases with each task, we update the same set of weights for the entire task sequence. Hence, SKILL and CoDeC target two different scenarios even though both utilize decentralized agents while continually learning a task sequence.

To compare SKILL with CoDeC, we employ SKILL in a scenario where each task's data is distributed across the agents. Similar to CoDeC, SKILL agents aim to learn a global generalized model with spatially and temporally distributed data. For iso-comparison, both SKILL and CoDeC use ResNet-18 pre-trained on ImageNet and perfect task oracle at test time. In SKILL, the agents do not communicate with their peers during training and share the bias parameters and heads only at the end of the training. Meanwhile, the agents in CoDeC communicate with their peers during training by utilizing the gossip averaging mechanism. Results in Table 12 show the importance of collaborative learning in scenarios where each task's data is spatially distributed. CoDeC achieves about 5.6% better accuracy than SKILL by the virtue of communicating while learning each task. As the number of agents increases from 8 to 16, SKILL performs much worse than CoDeC because the number of available training samples at each agent also reduces.

Table 12: Comparison between SKILL and CoDeC for Split miniImageNet over ResNet-18 using directed ring topology

| Agents | Setup | ACC (%) | BWT (%) |
|--------|-------|---------|---------|
| 8 | SKILL | 79.21 | 0.00 |
|   | CoDeC | 83.81 | 1.15 |
| 16 | SKILL | 74.33 | 0.00 |
|    | CoDeC | 80.94 | 0.77 |

Table 13: Results for non-IID data with 0.5 skew for Split CIFAR-100 and Split miniImageNet over a directed ring topology with 8 agents. D-STL is not a continual learning baseline and serves as an upper bound on performance.

| Dataset | Setup | ACC | BWT | CC |
|---------|-------|-----|-----|-----|
| Split CIFAR-100 | D-STL | $62.40 \pm 0.50$ | - | - |
|  | D-SI | $38.81 \pm 0.72$ | $-0.79 \pm 0.54$ | 1x |
|  | D-EWC | $49.20 \pm 0.35$ | $-0.13 \pm 0.21$ | 1x |
|  | CoDeC(f) | $50.71 \pm 0.47$ | $-0.46 \pm 0.23$ | 1x |
|  | CoDeC | $50.60 \pm 0.38$ | $-0.41 \pm 0.39$ | 2.53x |
| Split miniImageNet | D-STL | $60.89 \pm 0.61$ | - | - |
|  | D-SI | $45.64 \pm 0.51$ | $-1.97 \pm 0.32$ | 1x |
|  | D-EWC | $46.57 \pm 2.07$ | $-1.15 \pm 0.64$ | 1x |
|  | CoDeC(f) | $51.28 \pm 0.23$ | $-0.08 \pm 0.76$ | 1x |
|  | CoDeC | $50.93 \pm 0.16$ | $-0.88 \pm 0.50$ | 1.32x |

**Results on Non-IID Data:** We conducted experiments with 50% label-wise non-IIDness across the agents (i.e. skew=0.5) (Hsieh et al., 2019). CoDeC utilizes a variant of DPSGD (Lian et al., 2017) as the base decentralized algorithm for IID data across the agents. However, DPSGD has been shown to perform poorly with non-IID data. Techniques like Quasi-Global Momentum (QGM) (Lin et al., 2021) and Neighborhood Gradient Mean (NGM) (Aketi et al., 2023) were introduced to improve decentralized learning performance in the presence of data heterogeneity across the agents.

For our experiments, we employ QGM for learning with non-IID data as it improves performance without introducing extra communication overhead. Note that NGM performs better than QGM but at the cost of $2\times$ communication.

Our results in Table 13 show that CoDeC outperforms D-EWC and D-SI by 1.5% and 11.8% respectively for Split CIFAR-100. CoDeC achieves minimal backward transfer while achieving a CC of 2.53x. For Split miniImageNet, CoDeC achieves about 4-5% better accuracy than D-EWC and D-SI while incurring low BWT and CC of 1.32x.

**Runtime Comparison:** We present relative training runtimes in Table 14. For Split CIFAR-100 and Split miniImageNet, CoDeC(f), D-EWC and D-SI have similar runtimes. For 5-Datasets, D-EWC and D-SI have about 30% higher runtime than CoDeC(f).

CoDeC generally has the highest time due to the two extra matrix multiplications required for communication compression. Specifically, line 10 in algorithm 1 encodes model updates into coefficients which are communicated to the neighbors by each agent. Line 11 in algorithm 1 decodes the received coefficients back into model updates. Therefore, CoDeC achieves the highest accuracy and lossless communication compression at the cost of increased runtime.

Table 14: Training time (relative) for Split CIFAR-100, Split miniImageNet and 5-Datasets over a directed ring topology with 8 agents

| Dataset | Setup | Training time |
|---|---|---|
| Split CIFAR-100 | CoDeC(f) | 1 |
| | CoDeC | 1.27 |
| | D-EWC | 1.09 |
| | D-SI | 0.98 |
| Split miniImageNet | CoDeC(f) | 1 |
| | CoDeC | 1.68 |
| | D-EWC | 1 |
| | D-SI | 0.92 |
| 5-Datasets | CoDeC(f) | 1 |
| | CoDeC | 1.61 |
| | D-EWC | 1.30 |
| | D-SI | 1.31 |

**Algorithm 2** Decentralized Elastic Weight Consolidation (*D-EWC*)

---

**Input:** Each agent $i \in [1, N]$ initializes model parameters $\mathbf{x}_0^i$, step size $\eta$, mixing matrix $\mathbf{W} = [w_{ij}]_{i,j \in [1,N]}$, $\hat{\mathbf{x}}_{(0)}^i = 0$, $\mathbf{F}^l = [\ ]$ for all layers $l = 1, 2, ...L$, Fisher Matrix $\mathcal{F}^i = \{(\mathbf{F}^l)_{l=1}^L\}$, old model parameters $\mathbf{x}_{(0)}^i = 0$, $\mathcal{N}(i)$: neighbors of agent $i$ (including itself), $T$: total tasks, $K$: number of training iterations

Each agent simultaneously implements the TRAIN( ) procedure

1. **procedure** TRAIN( )
2.     **for** $\tau = 1, \ldots, T$ **do**
3.         **for** $k = 0, 1, \ldots, K-1$ **do**
4.             $d_{\tau,i} \sim \mathcal{D}_{\tau,i}$
5.             $\tilde{f}_{\tau,i}(d_{\tau,i}; \mathbf{x}_k^i) = f_{\tau,i}(d_{\tau,i}; \mathbf{x}_k^i) + \sum_{l=0}^L \frac{\lambda}{2} \mathbf{F}^l(\mathbf{x}_k^{i,l} - \mathbf{x}_{\tau-1}^{i,l})$
6.             $\mathbf{g}_k^i = \nabla \tilde{f}_{\tau,i}(d_{\tau,i}; \mathbf{x}_k^i)$
7.             $\mathbf{x}_{(k+\frac{1}{2})}^i = \mathbf{x}_k^i - \eta \mathbf{g}_k^i$
8.             $\mathbf{x}_{k+1}^i = \mathbf{x}_{(k+\frac{1}{2})}^i + \sum_{j \in \mathcal{N}(i)} w_{ij}(\hat{\mathbf{x}}_k^j - \mathbf{x}_k^i)$
9.             $\mathbf{q}_k^i = \mathbf{x}_{k+1}^i - \mathbf{x}_k^i$
10.            for each $j \in \mathcal{N}(i)$ **do**
11.                Send $\mathbf{q}_k^i$ and receive $\mathbf{q}_k^j$
12.                $\hat{\mathbf{x}}_{(k+1)}^j = \mathbf{q}_k^j + \hat{\mathbf{x}}_k^j$
13.            **end**
14.         **end**
15.         Save $\mathbf{x}_\tau^i$
16.         # EWC Update
17.         Update $\mathbf{F}^l$ for each layer $l$
18.         Update $\mathcal{F}^i = \{(\mathbf{F}^l)_{l=1}^L\}$
19.         $p = random(1, 2, ...N)$
20.         **if** $i == p$ **do**
21.           Gather $\mathcal{F}^i$ from all agents
22.           $\mathcal{F} = avg(\mathcal{F}^1, \mathcal{F}^2, ....\mathcal{F}^N)$
23.           Send $\mathcal{F}$ to all agents
24.         **end**
25.     **end**
26. **return**

**Algorithm 3** Decentralized Synaptic Intelligence (*D-SI*)

---

**Input:** Each agent $i \in [1, N]$ initializes model parameters $\mathbf{x}_0^i$, step size $\eta$, mixing matrix $\mathbf{W} = [w_{ij}]_{i,j \in [1,N]}$, $\hat{\mathbf{x}}_{(0)}^i = 0$, $\mathbf{\Omega}^l = [\ ]$ for all layers $l = 1, 2, ...L$, importance measure $\mathbf{\Omega}^i = \{(\mathbf{\Omega}^l)_{l=1}^L\}$, running importance estimate $\omega = [\ ]$, damping parameter $\xi = 10^{-3}$, old model parameters $\mathbf{x}_{(0)}^i = 0$, $\mathcal{N}(i)$: neighbors of agent $i$ (including itself), $T$: total tasks, $K$: number of training iterations

Each agent simultaneously implements the TRAIN( ) procedure

1. **procedure** TRAIN( )
2.     **for** $\tau = 1, \ldots, T$ **do**
3.         **for** $k = 0, 1, \ldots, K-1$ **do**
4.             $d_{\tau,i} \sim \mathcal{D}_{\tau,i}$
5.             $\tilde{f}_{\tau,i}(d_{\tau,i}; \mathbf{x}_k^i) = f_{\tau,i}(d_{\tau,i}; \mathbf{x}_k^i) + c\sum_{l=0}^L \mathbf{\Omega}^l(\mathbf{x}_k^{i,l} - \mathbf{x}_{\tau-1}^{i,l})^2$
6.             $\mathbf{g}_k^i = \nabla \tilde{f}_{\tau,i}(d_{\tau,i}; \mathbf{x}_k^i)$
7.             $\omega^l = \omega^l - \mathbf{g}_k^i * (\mathbf{x}_k^{i,l} - \mathbf{x}_{k-1}^{i,l})$
8.             $\mathbf{x}_{(k+\frac{1}{2})}^i = \mathbf{x}_k^i - \eta \mathbf{g}_k^i$
9.             $\mathbf{x}_{k+1}^i = \mathbf{x}_{(k+\frac{1}{2})}^i + \sum_{j \in \mathcal{N}(i)} w_{ij}(\hat{\mathbf{x}}_k^j - \mathbf{x}_k^i)$
10.            $\mathbf{q}_k^i = \mathbf{x}_{k+1}^i - \mathbf{x}_k^i$
11.            for each $j \in \mathcal{N}(i)$ **do**
12.                Send $\mathbf{q}_k^i$ and receive $\mathbf{q}_k^j$
13.                $\hat{\mathbf{x}}_{(k+1)}^j = \mathbf{q}_k^j + \hat{\mathbf{x}}_k^j$
14.            **end**
15.         **end**
16.         Save $\mathbf{x}_\tau^i$
17.         # SI Update
18.         $\mathbf{\Omega}^l = \mathbf{\Omega}^l + \frac{\omega^l}{(\mathbf{x}_\tau^{i,l} - \mathbf{x}_{\tau-1}^{i,l})^2 + \xi}$ (for each layer $l$)
19.         Update $\mathbf{\Omega}^i = \{(\mathbf{\Omega}^l)_{l=1}^L\}$
20.         $p = random(1, 2, ...N)$
21.         **if** $i == p$ **do**
22.           Gather $\mathbf{\Omega}^i$ from all agents
23.           $\mathbf{\Omega} = avg(\mathbf{\Omega}^1, \mathbf{\Omega}^2, ....\mathbf{\Omega}^N)$
24.           Send $\mathbf{\Omega}$ to all agents
25.         **end**
26.     **end**
27. **return**

