# OpenReview forum: "CoDeC: Communication-Efficient Decentralized Continual Learning"
_TMLR — Accepted by TMLR_

### Review · Reviewer_kjKk · 2023-11-08

**Summary Of Contributions:**

This paper proposes CoDeC, a novel communication-efficient decentralized continual learning algorithm that addresses these challenges. CoDeC mitigates catastrophic forgetting while learning a distributed task sequence by incorporating orthogonal gradient projection within a gossip-based decentralized learning algorithm. In addition, CoDeC includes a novel lossless communication compression scheme based on the gradient subspaces. The authors theoretically analyze the convergence rate for their algorithm and demonstrate through an extensive set of experiments that CoDeC successfully learns distributed continual tasks with minimal forgetting. The proposed compression scheme results in up to 4.8× reduction in communication costs without any loss in performance.

**Audience:**

Yes

**Broader Impact Concerns:**

I do not have ethical concerns.

**Claims And Evidence:**

Yes

**Requested Changes:**

Please see the review on weaknesses above.

**Strengths And Weaknesses:**

**Strengths:**

1.	The considered problem, communication-efficient decentralized continual learning, is very important and well-motivated in the distributed learning and continual learning literatures.
2.	The authors provide theoretical guarantees for the convergence rate of the proposed algorithm.
3.	The authors conduct experiments on a wide range of datasets and evaluate the proposed algorithm under various performance metrics, including accuracy, backward transfer and communication compression.

**Weaknesses:**

1.	The provided theoretical results seem to only provide the convergence rate to the stationary point? Does the proposed decentralized continual learning scheme converge to the global optima? The authors should give more discussion on the convergence guarantee of algorithm CoDeC.
2.	How can we see the performance of algorithm CoDeC in terms of communication efficiency from the provided theoretical results? Is there any theoretical guarantee for communication costs?
3.	How do the provided theoretical results reflect that CoDeC works well in avoiding catastrophic forgetting? It would be better if the authors can give more discussion on this.

---

> ### Author Response · Authors · 2023-11-20
> **Reply to Reviewer kjKk**
>
> We would first like to thank the reviewer for the feedback. We answer the questions raised in the weakness section here:
>
> 1. For convex loss functions, it is possible to provide convergence guarantees for Stochastic Gradient Descent (SGD) to a global minima as shown in [1]. However, authors in [1] also establish that SGD for non-convex and L-smooth loss functions are proven to converge only to first-order stationary points. More recent works like [2] proposed variants of SGD, which under stricter assumptions of smooth hessians are proven to escape saddle points and converge towards a local minima.
> Our theoretical results are applicable for non-convex loss functions which are L-smooth (i.e. assumption 1 in the main paper), similar to [1]. This assumption is also utilized in other decentralized learning algorithms DPSGD [3] and CGA [4]. Consequently, our convergence guarantees align with this previous line of works in the decentralized learning paradigm. Theorem 2 characterizes the convergence of $\bar{\mathbf{x}}_{k}$, which is the average of all local models $\mathbf{x}_k^i$ (at training iteration $k$). To prove this, we bound the norm of the gradient for the global loss function $\mathcal{F}(\bar{\mathbf{x}})$ over the average model $\bar{\mathbf{x}}$. The result of theorem 2 and corollary 3 proves that this gradient norm converges to 0 at the rate of $O(\frac{1}{\sqrt{NK}})$, which is similar to DPSGD [3], a state-of-the-art decentralized learning algorithm.
>
> 2. The communication efficiency of CoDeC comes from the fact that the gradient updates are constrained to lie in Residual Gradient Space (RGS). The threshold $\epsilon_{th}$ is a hyperparameter which is tuned to mitigate catastrophic forgetting, and it lies between $0< \epsilon_{th}<1$. This ensures that for a given task, some (but not all) basis directions are added to CGS matrix $\textbf{M}$, which in turn implies that the dimension of the subspace RGS is smaller than the entire gradient space. Hence, reduction in the communication cost is guaranteed in our setup by the virtue of taking orthogonal gradient updates to avoid forgetting. The exact reduction for each task depends upon the number of basis vectors added to CGS matrix $\textbf{M}$ (and subsequently removed from the RGS matrix $\textbf{O}$) to mitigate forgetting. This is determined by the choice of $\epsilon_{th}$ for a given model and dataset.
>  Moreover, we ensure that all the agents have the same $\textbf{M}$ and $\textbf{O}$ matrices (section 3.3). Hence, the reconstruction of received coefficients $\mathbf{c}^i_k$ is exact, resulting in a lossless compression scheme. This implies that CoDeC and CoDeC(f) have the same rate of convergence, as shown in Figure 7 (Appendix).
>
> 3. Our theoretical analysis focuses on the convergence analysis of continual learning techniques like GPM in decentralized learning scenarios. In a single-agent centralized setting, GPM has been shown to effectively alleviates forgetting by utilizing orthogonal gradient updates and ensuring that the updates lie in RGS. CoDeC adopts this property and ensures that every update on every agent lies in RGS. CoDeC has two update steps that modify the model parameters: (1) Local orthogonal gradient update: $\mathbf{x}_{(k+\frac{1}{2})}^i=\mathbf{x}\_k^{i} - \eta \tilde {\mathbf{g}}^i\_k$ and (2) Gossip averaging mechanism: $\mathbf{x}\_{k+1}^i =\mathbf{x}\_{(k+\frac{1}{2})}^i+\sum\_{j  \in \mathcal{N}(i)} w\_{ij} (\hat{\mathbf{x}}\_k^{j}-\mathbf{x}\_k^{i})$.
>
>     (1) is the same as GPM’s orthogonal gradient update which has been shown to avoid forgetting. We claim that the gossip update step in (2) also lies in RGS and thus avoids catastrophic forgetting. We prove this using the linearity property of the vector spaces and induction.  Say, $\alpha\_k^i = \sum\_{j} w\_{ij} (\hat{x}\_k^j - x\_k^i)= \sum\_{j} w\_{ij} (x\_k^j - x\_k^i)$.  $\alpha_0^i$ lies in RGS as $x_0^j=x_0^i$ (synchronized initialization). Now, $\alpha_{k+1}^i=\sum_{j} w_{ij} (\alpha\_k^j - \eta (\tilde{g}^j_k-\tilde{g}^i_k))$. GPM update ensures that gradients $\tilde{g}^{j}\_k$ and $\tilde{g}^{i}\_k$ lie in RGS and from induction, we have $\alpha\_k^j$ to lie in RGS. Hence, from linearity, we conclude that $\alpha\_{k+1}^i$ lies in RGS for any agent $i$.
>
>    We will refine the theoretical analysis by incorporating this as a lemma and update the paper.
>
> [1] Ghadimi, Saeed et al. Stochastic first-and zeroth-order methods for nonconvex stochastic programming. SIAM Journal on Optimization, 2013.
> [2] Ge, Rong et al. Escaping from saddle points—online stochastic gradient for tensor decomposition. In Conference on Computational Learning Theory, 2015.
> [3] Lian, Xiangru et al. Can decentralized algorithms outperform centralized algorithms? a case study for decentralized parallel stochastic gradient descent, 2017.
> [4] Esfandiari, Yasaman et al. Cross-gradient aggregation for decentralized learning from non-iid data. CoRR, abs/2103.02051, 2021.

---

### Review · Reviewer_m7Vb · 2023-12-02

**Summary Of Contributions:**

The paper proposes a novel algorithm for decentralized continual learning called CoDeC which uses Gradient Projection Memory (GPM) to prevent catastrophic forgetting in decentralized learning settings where data from different tasks is presented sequentially. The algorithm also reduces the communication cost by sending GPM coefficients of the model updates instead of the updates themselves. Experiments show accuracy improvements over baselines and upto 4.8x reduction in communication costs for a range of datasets and network topologies.

**Audience:**

Yes

**Claims And Evidence:**

Yes

**Requested Changes:**

1. It appears from Section 2.3 that SKILL is the closest related work to CoDeC and as such I feel it should be included as a baseline if possible. Even if it learns the tasks independently and not collaboratively it should be possible to compare the final task level accuracy of SKILL and CoDeC as well as the communication cost of the two appraoches right? If it is not possible please clarify why that is the case.

2. It is not clear from Lines 8 and 9 of Algorithm 1 why $\mathbf{q}_{i}^{k}$ lies in the RGS subspace. Please clarify.

3. From line 14 it seems like $\hat{x_{0}}^{j}$ will need to be communicated to all nodes since we don't have any coefficients at the beginning to reconstruct it, unless $\hat{x}_{0}^{j}$ is the same for all $j$. Please clarify if that is the case.

 4. It is not clear to me how you ensure that the CGS vectors being added to the GPM matrix in round $\tau$ are orthogonal to the CGS vectors added in previous rounds. Please explain this more clearly.

5. Do you have any intuition/explanations for the mixed BWT results in Table 3 and 4? If yes, please include a discussion of the same in the paper.

**Strengths And Weaknesses:**

Strengths:

1. The paper combines gossip averaging and GPM to propose a novel algorithm for decentralized continual learning. This appears to be the first work that has looked at continual learning in the decentralized (non-federated) setting and as such it addresses an important research question. The algorithm is also backed by both theoretical analysis and empirical improvements in accuracy over baselines.

2. The authors also provide a novel approach for communication compression in this setting by communicating residual coefficients whose dimensionality is much smaller than that of the gradient. This provides upto 4.8x reduction in communication cost in experiments.

3. The paper also introduces a new dataset MedMNIST-5 which can serve as a benchmark biomedical dataset for future distributed continual learning works.

Weaknesses:

1. The paper makes some strong assumptions like IID data at each node and presentation of tasks at the same order at all nodes, especially because these assumptions are more likely to be violated in continual learning settings where data at each node may have its own spatial/temporal distribution and it's not necessary that the data for the same task is collected at all nodes at the same time.

2. Some parts of the algorithm are not clearly explained. Please see requested changes below for suggestions for improvement.

3. SKILL, which according to Section 2.3 is the only other related work studying decentralized continual learning for neural networks, is not included as a baseline.

4. The performance of CoDeC w.r.t the BWT metric appears to be mixed in Table 3 and 4. Since the main goal of continual learning is to prevent forgetting it appears that BWT is a more important metric than ACC and so these mixed results seem to suggest that CoDeC isn't really meeting its main goal for some settings at least.

---

> ### Author Response · Authors · 2023-12-20
> **Reply to Reviewer m7Vb**
>
> We thank the reviewer for their valuable time and feedback. We provide clarifications for the points raised in requested changes:
> 1. Each agent in SKILL uses a common pre-trained frozen backbone built-in at initialization so that only the last layer (or head) and a unique set of bias parameters are learned for each task.  After training, these bias parameters and task-specific heads are shared among agents via a fully connected graph. However, each SKILL agent is independently learning a different task at a given time, lacking the concept of collaborative learning as demonstrated in CoDeC. The data for each task is spatially distributed in our setup, while SKILL assumes access to the entire data available in a centralized manner at a single agent. Hence, in CoDeC it becomes essential to communicate with peers during training. Moreover, we do not assume access to a pre-trained model at initialization, and update the weights for each task while mitigating forgetting. Unlike SKILL where the number of learned parameters increases with each subsequent task, we update the same set of weights for the entire task sequence. Hence, SKILL and CoDeC target two different scenarios even though both utilize decentralized agents while continually learning a task sequence.
>
>    To compare SKILL with CoDeC, we employ SKILL in a scenario where each task's data is distributed across the agents. Similar to CoDeC, SKILL agents aim to learn a global generalized model with spatially and temporally distributed data. For iso-comparison, both SKILL and CoDeC use ResNet-18 pre-trained on ImageNet and perfect task oracle at test time. In SKILL, the agents do not communicate with their peers during training and share the bias parameters and heads only at the end of the training. Meanwhile, the agents in CoDeC communicate during training by utilizing the gossip averaging mechanism. Results in Table 1 show the importance of collaborative learning in scenarios where each task's data is spatially distributed. CoDeC achieves $5.6\\%$ better accuracy than SKILL by the virtue of communicating while learning each task. As the number of agents increases from 8 to 16, SKILL performs much worse than CoDeC because the number of training samples at each agent reduces.
>
> \begin{array}{ | c | c | c | c |}
> \hline
> \text{Agents} & \text{Setup} & \text{ACC\(\\%\)} & \text{BWT\(\\%\)} \\\\
> \hline
> 8 & \text{SKILL} & \text{79.21} & 0.00 \\\\
> & \text{CoDeC} & 83.81 & 1.15 \\\\
> \hline
> 16 & \text{SKILL} & 74.33 & 0.00 \\\\
> & \text{CoDeC} & 80.94 & 0.77 \\\\
> \hline
> \end{array}
> Table 1: Comparison between SKILL and CoDeC for Split miniImageNet over pretrained ResNet-18 using directed ring
>
>    We will include this discussion and the subsequent results in the Appendix in the revised version.
>
> 2. Line 8: $\mathbf{x}\_{k+1}^{i}=\mathbf{x}\_{(k+\frac{1}{2})}^i+ \sum\_{j \in \mathcal{N}(i)} w\_{ij}(\hat{\mathbf{x}}\_k^{j}-\mathbf{x}\_k^{i})$
>
>     Line 9: $\mathbf{q}\_{k}^i= \mathbf{x}^i\_{k+1}-\mathbf{x}^i\_{k}$
>
>      After simplifying $\mathbf{q}\_{k}^i$:
>      \begin{equation}
>      \mathbf{q}\_{k}^i= \mathbf{x}\_{(k+\frac{1}{2})}^i+ \sum\_{j \in \mathcal{N}(i)} w\_{ij}(\hat{\mathbf{x}}\_k^{j}-\mathbf{x}\_k^{i})-
>      \mathbf{x}^i\_{k}     \hspace{20mm}      (1)
>      \end{equation}
>
>      From line 7 in Algorithm 1 we have, $\mathbf{x}\_{(k+\frac{1}{2})}^i=\mathbf{x}\_k^{i}- \eta \tilde{\mathbf{g}}^i\_{k} $
>
>     Substituting $\mathbf{x}\_{(k+\frac{1}{2})}^i$ in equation 1,
>     $\mathbf{q}\_{k}^i= - \eta \tilde{\mathbf{g}}^i\_{k} + \sum\_{j \in \mathcal{N}(i)} w\_{ij}(\hat{\mathbf{x}}\_k^{j}-\mathbf{x}\_k^{i})$
>
>    Now, we know that $\tilde{\mathbf{g}}^i_{k}$ is the orthogonal gradient component of $\mathbf{g}^i_{k}$, and hence it lies in RGS subspace. If the gossip update component i.e. $\sum\_{j \in \mathcal{N}(i)} w\_{ij}(\hat{\mathbf{x}}\_k^{j}-\mathbf{x}\_k^{i})$ also lies in RGS, we can conclude that $\mathbf{q}\_{k}^i$ lies in RGS by linearity.
>
>    We prove this using the linearity property of the vector spaces and induction.
>
>    Say $\alpha\_k^i = \sum\_{j} w\_{ij} (\hat{x}\_k^j - x\_k^i)= \sum\_{j} w\_{ij} (x\_k^j - x\_k^i)$. $\alpha\_0^i$ lies in RGS as $x\_0^j=x\_0^i$ (synchronized initialization).
>
>    Now, $\alpha\_{k+1}^i=\sum\_{j} w\_{ij} (\alpha\_k^j - \eta (\tilde{g}^j\_k-\tilde{g}^i\_k))$. GPM update ensures that gradients $\tilde{g}^{j}\_k$ and $\tilde{g}^{i}\_k$ lie in RGS and from induction, we have $\alpha\_k^j$ to lie in RGS. Hence, from linearity, we conclude that $\alpha\_{k+1}^i$ lies in RGS for any agent $i$.
>
>    We will include this as a lemma in the revised version of the paper.
>
> 3. We clarify in section 3.2 that all hyperparameters are synchronized between the agents at the beginning of the training. Hence, $\mathbf{x}\_0^{i}= \mathbf{x}\_0^{j}$. Also, as mentioned in algorithm 1, $\hat{\mathbf{x}}\_{0}^i=\mathbf{x}\_0^{i}$. Therefore, $\hat{\mathbf{x}}\_0^j= \mathbf{\hat{\mathbf{x}}}\_0^i$ for all $i$.

---

> ### Author Response · Authors · 2023-12-20
> **Continued reply to Reviewer m7Vb**
>
> 4. For task $\tau > 1$, we ensure that the CGS vectors being added to the GPM matrix in round $\tau$ are orthogonal to all the CGS vectors already stored in $\mathbf{M}$. Before performing SVD on the representation matrix $\mathbf{R}\_\tau^l$ for each layer $l$, we perform the following projection step:
>
>     \begin{equation}
>         \hat{\mathbf{R}}\_\tau^l= \mathbf{R}\_\tau^l - ({\mathbf{M}^l}{(\mathbf{M}^l)}^T)\mathbf{R}\_\tau^l
>     \end{equation}
>
>    SVD is then performed on $\hat{\mathbf{R}}_\tau^l$ and new orthogonal basis vectors are added to $\textbf{M}$. This ensures that the newly added basis vectors for task $\tau$ are unique and orthogonal to the vectors already present in $\textbf{M}$.
>
>     We thank the reviewer for pointing this out. We will update the paper by including this clarification in Section 3.3.
>
> 5. Table 3, 4 and 5 correspond to results for datasets Split CIFAR-100, miniImageNet and 5-Datasets respectively. The reported BWT for CoDeC is relatively higher for 5-Datasets in Table 5. However, CoDeC outperforms the baselines by upto $5\\%$ for this case. We believe that final average accuracy (ACC) and backward transfer (BWT) are both equally important in continual learning scenarios. It is possible to achieve zero BWT by freezing the model weights after learning the first task. However, that leads to sub-optimal ACC due to lack of ability to learn the subsequent tasks effectively. In our setup, we strive to achieve a balance between these two metrics by tuning the threshold hyperparameter $\epsilon\_{th}$ accordingly. For 5-Datasets, it is possible to achieve a lower BWT, but at the cost of lower ACC. The results presented in Table 5 are for $\epsilon\_{th}=0.965$. We conduct an additional experiment by utilizing a higher $\epsilon\_{th}$ for 16 agents connected in the ring topology:
>
>    \begin{array}{ |c| c |c|}
>      \hline
>       \text{Setup} & \text{ACC (\\%)} & \text{BWT (\\%)} \\\\
>       \hline
>       \text{CoDeC (table 5 in the paper)} & 86.41 & -4.37 \\\\
>       \text{CoDeC with $\epsilon\_{th}=0.99$} & 85.20 & -1.10 \\\\
>       \hline
>    \end{array}
>
>    These results demonstrate that a lower $\epsilon\_{th}$ can lead to higher ACC but higher BWT as well. Also, a higher $\epsilon\_{th}$ leads to lower BWT but potentially lower ACC. Hence, it is possible to tune this hyperparameter to achieve lower BWT or higher ACC as required.
>
>    We will update the paper by including this discussion in Section 6 under performance comparison.
>
>
> Now, we proceed to address the mentioned weaknesses:
> 1. We thank the reviewer for pointing this out. Even though our setup assumes IID data distribution across the agents, we believe it is possible to extend our approach to scenarios where the data distribution is non-IID. This is an active area of research in federated learning as well as decentralized learning communities. We utilize a variant of DPSGD [1] in our work, but it has been shown to perform poorly with non-IID data. Techniques like QGM [2] and NGM [3] can be utilized to improve decentralized learning performance in the presence of data heterogeneity across the agents. We will update the paper and include this in a new section titled "Limitations and Future Work".
>
>     A practical real-world application of our setup is when multiple healthcare organizations aim to learn
>     a global generalized model without sharing the locally accessible patients’ data (as mentioned in Section 1).  These models need to be updated with the emergence of variants of a disease, new diseases, or new diagnostic methods in a continual manner. Hence, we extended our analysis and included results on a new decentralized continual learning benchmark MedMNIST-5 (Table 6). Our setup is generally applicable in scenarios where the data evolves with time and all agents are interested in learning a global generalized model but do not have access to the entire data for each task.
>
>
> (2,3 and 4 have been covered through our clarifications on the requested changes.)
>
> [1] Lian, Xiangru et al. Can decentralized algorithms outperform centralized algorithms? a case study for decentralized parallel stochastic gradient descent, NeurIPS 2017.
>
> [2] Lin, Tao et al. "Quasi-Global Momentum: Accelerating Decentralized Deep Learning on Heterogeneous Data", ICML 2021.
>
> [3] Aketi S A et al. "Neighborhood Gradient Mean: An Efficient Decentralized Learning Method for Non-IID Data", TMLR 2023.

---

### Review · Reviewer_Gvsm · 2023-12-30

**Summary Of Contributions:**

In this paper, the authors present CoDeC, a method for decentralized continual learning. The method utilizes orthogonal gradient projection memory, in which each layer of the network is partitioned into orthogonal subspaces with the aim being to mitigate interference between previous tasks and new updates.  Decentralized learning is able to be performed by a gossip-based averaging  in which model parameters are first communicated to neighbors, and averaged locally. Neighboring model updates are then computed by only sending coefficients of the bases spanning the RGS subspace from the orthogonal gradient projection.

The authors provide theoretical and empirical analysis of convergence, as well as empirical comparisons of the performance of their method with baselines which are reasonable given the unique problem setting and focus on continual learning.

**Audience:**

Yes

**Claims And Evidence:**

No

**Requested Changes:**

1. Provide runtime analysis and comparisons with other methods.
2. Provide technical specifications for the implementation details.

**Strengths And Weaknesses:**

## Strengths
1. The writing is generally clear - I have no significant issues with the presentation from a mechanical perspective and any copyediting mistakes do not affect overall clarity.
2. As far as I am aware, this use of GPM in decentralized setting is unique, and the problem of continual learning in decentralized settings is important.
3. The empirical results which are presented are generally convincing, and -  combined with the results in the appendix - thorough.
## Weaknesses
1. The greatest weakness in my opinion is that the proposed approach is not fit for the intended use. The paper claims enabling decentralized continual learning and the abstract even describes a typical decentralized setup with private data at different sites. However, this setup is very difficult to imagine as having identically distributed data. In fact, the majority of the cases where private datasets are virtually pooled to obtain a stronger model are motivated by enriching the overall data distribution by combining quite diverse parts of the overall joint spread across the sites. The functioning and assumptions of the algorithm and, I assume, the proof will not apply to realistic situations, where data across sites has different distributions and covers different regions of the domain. Additionally, it is not reasonable to expect tasks to perfectly align across the sites in the order that they will be presented to the model. Moreover, there is no reason to expect the bases computed via an SVD at each site to be the same or even similar. This situation is not clearly explained in the paper (see my questions in the corresponding section).
2. A possible explanation of the weakness number 1, is that the method is for decentralized training on a local server across multiple nodes. In this case, the users will have full control of reshuffling the data and splitting it across the nodes according to any assumptions needed by the algorith. However, in this case the complete omission of runtime analysis becomes the major concern. Although I do acknowledge that communication limitations are important in an of themselves, it is difficult to argue for the benefits of communication compression alone without a comparable analysis of runtime. It would be straightforward to provide both a theoretical and empirical analysis of runtime, and I think this is a necessary inclusion. Even if the runtime analysis results in slower runtime than baselines, the authors are responsible for highlighting this limitation (or strength if the method is faster) and arguing with respect to these findings.
   - Note, runtime analysis and comparison with simpler pre-exisiting baselines is an important omission in either the cases of training on decentralized private data or the data deliberately distributed across nodes in a single server.
3. A related, but somewhat  lesser, exclusion is that no theoretical analysis of the communication is provided (as far as I have found).  It is standard in this field to provide a theoretical analysis of the communication complexity, and this would help to augment the discussion in which the authors discuss their compression ratio.
4. A further weakness is that the authors omit a number of important implementation details, such as the hardware information both for individual agents (CPU, GPU, etc) and the network (connection bandwidth, etc). These factors are usually most important when runtime is discussed; however, network bandwidth can also be important for arguing for the benefits of compression (there are arguments that compression is not always needed in high-bandwidth scenarios, etc).
## Questions
1. My main question regards the first distributed communication step (algoritm 1, line 8 ) in which parameters are communicated across the network.  It seems to me that this step could be skipped entirely if we start with the same model parameters at each agent. This is a common practice in distributed SGD for example, where we assume that models start with the same set of parameters, and so we only require gradient updates in order to arrive at a common model. Why is it desirable to assume differently initialized parameters across agents? Couldn't we avoid this unnecessary overhead by assuming a common starting seed?

---

> ### Author Response · Authors · 2024-01-08
> **Reply to Reviewer Gvsm**
>
> We thank the reviewer for their time and feedback. We start by addressing the points raised in the weaknesses section:
>
> **1 & 2:** *IID data assumption*: Even though our setup assumes IID data distribution across the agents, our approach can be extended to scenarios where the data distribution is non-IID. We provide additional results for non-IID data distributions across the agents in Table 13 in the revised version of the paper.
>
> Note that enabling decentralized learning in the presence of data heterogeneity is an active area of research [2,3,5], and is a challenging problem to tackle even without communication compression. In fact, current state of the art works like NGM [3] and CGA [5] require an additional round of communication to achieve good performance. Moreover, IID data distribution is a common assumption in most communication compression works for decentralized learning [4].
>
> *perfectly aligned tasks*: A practical real-world application of our setup is when multiple healthcare organizations aim to learn a global generalized model without sharing the locally accessible patients’ data (as mentioned in Section 1). These models need to be updated with the emergence of variants of a disease, new diseases, or new diagnostic methods in a continual manner. Our setup is generally applicable in scenarios where the data evolves with time and all agents are interested in learning a global generalized model but do not have access to the entire data for each task.
>
> *runtime analysis*: We provide the runtime analysis in Table 14 in the revised paper.
>
> *SVD computed bases*: As mentioned at the end of section 3.2, SVD is calculated using a subset of training data at one randomly chosen agent and communicated to other agents iteratively using the communication graph. This is valid because of our IID data assumption, and all agents converging to the same model at the end of a training round. In this manner, the CGS and RGS matrices at each agent are the same. On the other hand, for results with non-IID data in Table 13 we calculate SVD at every agent and extract a union of the resulting basis vectors to store in the CGS and RGS matrices.
>
> *data reshuffling*: Our method is intended for training at the edge, where each agent has its own private data. Hence, we do not make any such assumptions about the possibility of data reshuffling across the agents due to privacy concerns. Agents in the same location and similar interests may have homogeneous (i.e. IID) data distributions. In such a scenario, the class distribution across the agents is same but the training samples at each agent are still unique. To emulate this scenario, we distribute the training samples/tasks in an IID manner across the agents without any data overlap. For instance, for a graph size of 4 agents, each agent has 5000/4 = 1250 training samples for a particular task in Split CIFAR-100 (mentioned in A.8 in the Appendix).
>
> **3:** The communication compression presented in CoDeC is lossless, and the communication efficiency comes from the fact that the gradient updates are constrained to lie in Residual Gradient Space (RGS). The degree of compression depends upon $\epsilon_{th}$, a hyperparameter that is tuned to mitigate forgetting. The exact reduction for each task depends upon the number of basis vectors added to CGS matrix $\mathbf{M}$ (and subsequently removed from the RGS matrix $\mathbf{O}$).
>
> In our convergence analysis, we included $\mu$, which signifies how constrained the gradient space is. Subsequently, Corollary 3 shows that $\mu$ does not affect the rate of convergence, proving that both CoDeC and CoDeC(f) converge at the same rate. This is also demonstrated through the experimental results in Figure 7 in the Appendix.
>
> **4:** We provide the requested details in the first paragraph in Appendix in the revised version. Our work focuses on on-device training scenarios at the edge, where network bandwidth is generally limited and communication compression becomes a necessity.

---

> ### Author Response · Authors · 2024-01-08
> **Continued reply to Reviewer Gvsm**
>
> Clarification for **Question 1:**
>
> We don't assume differently initialized parameters across agents.
> It is clearly stated in Section 3.2 that all agents start with a common model and all the hyperparameters are also synchronized across the agents at the beginning of the training.
> Each agent computes model updates ($\mathbf{q}\_{k}^i= \mathbf{x}^i\_{k+1}-\mathbf{x}^i\_{k}$) and communicates the associated coefficients with its neighbors as shown in line 12, algorithm 1. We would like to emphasize that we use a decentralized learning setup [1] with no central server which is different from federated (centralized) learning and distributed SGD.
>
> Line 8 in algorithm 1 is the gossip averaging step commonly used in decentralized learning algorithms [1-5]. Unlike distributed SGD where all agents communicate in a fully connected topology through mechanisms like all-reduce, decentralized learning focuses on sparsely connected topologies like ring and torus. In such a setup, even when all agents start with a common model, they have different models at a given time during the training. Communicating only the local gradient updates does not guarantee convergence. Therefore, communicating the model parameters or the model updates and accumulating these through gossip averaging (line 8) is crucial for convergence in decentralized learning.
>
> For the **requested changes:**
>
> 1. Please refer to Table 14 in the revised version for the requested analysis and comparison.
>
> 2. We perform our experiments on a single machine with 4 NVIDIA GeForce GTX 1080 Ti GPUs. All the agents in our experiments are distributed evenly over these 4 GPUs. For instance, in the case of a 16-agent ring/torus topology, each GPU is utilized by 4 agents.
> In the revised version of the paper, we include this in the Appendix (first paragraph).
>
> References:
>
> [1] Lian, Xiangru et al. Can decentralized algorithms outperform centralized algorithms? a case study for decentralized parallel stochastic gradient descent, NeurIPS 2017.
>
> [2] Lin, Tao et al. "Quasi-Global Momentum: Accelerating Decentralized Deep Learning on Heterogeneous Data", ICML 2021.
>
> [3] Aketi S A et al. "Neighborhood Gradient Mean: An Efficient Decentralized Learning Method for Non-IID Data", TMLR 2023.
>
> [4] Koloskova, Anastasia et al. "Decentralized Deep Learning with Arbitrary Communication Compression", ICLR 2020.
>
> [5] Esfandiari, Yasaman, et al. "Cross-gradient aggregation for decentralized learning from non-iid data." International Conference on Machine Learning. PMLR, 2021.

---

### Author Response · Authors · 2024-01-08
**Paper Revision Based on Suggested Changes**

We have revised the paper based on the reviewers' suggestions and highlighted the added content with blue color. Specifically, this revised version includes the following:
1. Results on non-IID data distribution across the agents (Table 13)
2. Comparison between SKILL and CoDeC (Table 12)
3. Results on runtime comparisons (Table 14)
4. Hardware implementation details (Appendix)
5. Discussion on mixed BWT results for 5-Datasets (Section 6)
6. Proof for the update step in line 9 ($\mathbf{q_k^i}$) lying in RGS (Appendix A.6)
7. Details about how vectors added to the CGS matrix are orthogonal to the existing vectors (Section 3.3)

We have addressed all the concerns raised by the reviewers and would be happy to clarify any further questions.

---

### Decision · Action_Editor_zCAe · 2024-03-09

**Recommendation:** Accept as is

**Comment:**

The main concern raised by the reviewers is the unrealistic IID assumption, which will not be verified in practice. The authors discussed this in the revised version and show encouraging new experimental results, but the concern remains. While I share the concern that this work might not be effective in practice, the proposed method is sound and algorithms appear correct. In line with TMLR's goals, I am thus in favour of accepting the work.

**Audience:**

Continual and decentralised learning are topics of interest the community. The problems tackled in this paper are important, even though  one might questions some of the assumptions made by the authors, and reported results worth sharing with the community.

**Claims And Evidence:**

Aside from the practical applicability of the method, there was a consensus among the reviewers that the claims made by the authors were supported. The authors provided additional evidence to address the questions and concerns raised by the reviewers and all reviewers were satisfied with the answers and additional material provided, as well as the additional experiments conducted (e.g., non IID results, comparison between SKILL and CoDeC, runtime comparisons, hardware details, etc.). The only outstanding discussion point is the validity of the IID assumption in practice, which all reviewers considered to be unrealistic.